# Deep RL Needs Deep Behavior Analysis: Exploring Implicit Planning by Model-Free Agents in Open-Ended Environments

Riley Simmons-Edler[1,2]*    Ryan P. Badman[1,2]*    Felix Baastad Berg[3]    Raymond Chua[4]

John J. Vastola[1,2]    Joshua Lunger[5]    William Qian[6,2]    Kanaka Rajan[1,2]†

[1]Department of Neurobiology, Harvard Medical School
[2]Kempner Institute for the Study of Natural and Artificial Intelligence, Harvard University
[3]Department of Mathematics, NTNU
[4]School of Computer Science, McGill University & Mila
[5]Department of Computer Science, University of Toronto
[6]Biophysics Graduate Program, Harvard University
{riley_simmons-edler, ryan_badman, kanaka_rajan}@hms.harvard.edu

## Abstract

Understanding the behavior of deep reinforcement learning (DRL) agents—particularly as task and agent sophistication increase—requires more than simple comparison of reward curves, yet standard methods for behavioral analysis remain underdeveloped in DRL. We apply tools from neuroscience and ethology to study DRL agents in a novel, complex, partially observable environment, ForageWorld, designed to capture key aspects of real-world animal foraging—including sparse, depleting resource patches, predator threats, and spatially extended arenas. We use this environment as a platform for applying joint behavioral and neural analysis to agents, revealing detailed, quantitatively grounded insights into agent strategies, memory, and planning. Contrary to common assumptions, we find that model-free RNN-based DRL agents can exhibit structured, planning-like behavior purely through emergent dynamics—without requiring explicit memory modules or world models. Our results show that studying DRL agents like animals—analyzing them with neuroethology-inspired tools that reveal structure in both behavior and neural dynamics—uncovers rich structure in their learning dynamics that would otherwise remain invisible. We distill these tools into a general analysis framework linking core behavioral and representational features to diagnostic methods, which can be reused for a wide range of tasks and agents. As agents grow more complex and autonomous, bridging neuroscience, cognitive science, and AI will be essential—not just for understanding their behavior, but for ensuring safe alignment and maximizing desirable behaviors that are hard to measure via reward. We show how this can be done by drawing on lessons from how biological intelligence is studied.

---

*Equal contributions
†Corresponding author

39th Conference on Neural Information Processing Systems (NeurIPS 2025).

# 1 Introduction

As reinforcement learning (RL) researchers pursue more complex, naturalistic, and open-ended tasks [1], there is a growing need for tools to understand both the detailed behaviors and the neural dynamics of deep RL agents. This challenge mirrors long-standing problems in neuroscience, where researchers have developed extensive methods for joint behavioral and neural analysis in animals to understand biological intelligence. While machine learning (ML) has adopted neural representation analysis tools from neuroscience, mature behavioral analysis methods from neuroscience, cognitive science, and ethology [2, 3] remain underused in DRL. Open-ended RL tasks typically involve partial observability, memory, planning, and spatial reasoning—yet we still lack a systematic understanding of the strategies agents use to solve them [4–9]. Further, despite thousands of benchmarks—from video games [10] to 2D/3D gridworlds [11, 12]—there are no standard methods for analyzing or comparing agent behavior across tasks. RL evaluation still focuses primarily on reward curves and aggregate performance, which, while useful, offer limited insight into *how* agents solve structured tasks. The lack of behavioral analysis is a key bottleneck—limiting insight into why agents succeed or fail. Without understanding behavior, it's difficult to diagnose algorithmic performance. Applied RL also stands to benefit from removing this bottleneck: more complex tasks require both better agent planning and deeper diagnostics to verify expected behavior [13–15].

Notably, this is not just a machine learning problem—interpreting behavior in partially observable, reward-driven environments is also a core challenge in neuroscience and cognitive science. Inspired by how animals are studied in neuroscience and ethology, we propose a behavior-first approach to analyzing DRL agents. This approach enables richer insight into both learned behavior and internal representations, drawing on joint behavioral-neural analysis tools that have proven essential for understanding biological intelligence. We ground this proposal in **ForageWorld**, a novel task suite in which agents forage and survive in procedurally generated, large-scale, partially observable environments with temporally extended dynamics. These pressures reflect the ecological tradeoffs animals encounter during natural foraging [16, 17], and align with calls to push foraging tasks in neuroAI beyond the traditional bandit paradigms [18–23]. ForageWorld builds on Craftax [12], combining complex naturalistic environment features with efficient training. We extend the GPU-accelerated Craftax with features such as patchy resources and structured threats to better capture key elements of animal foraging. These additions place strong demands on memory, spatial reasoning, and long-term planning, presenting a non-trivial challenge for modern DRL agents. In addition, foraging serves as a useful metaphor for core robotics tasks such as navigation, cleaning, and search and rescue [24], and underlies many popular RL benchmarks, including Minecraft and NetHack [25].

To support behavior-first evaluation in DRL, we introduce an analysis framework (Table 1) that maps key behavioral and neural targets to diagnostic methods. This framework is showcased by the results in this work and is reusable across neuroAI, interpretability, and broader RL applications.

**Our key contributions are:**
(1) We develop **ForageWorld**, a naturalistic RL task suite with ecological structure—such as depleting resources, predators, partial observability, and open-ended survival goals—to study planning and memory in model-free agents.
(2) We adapt tools from neuroscience and ethology—including path analysis, generalized linear models (GLMs), and recurrent state decoding—for **joint behavioral and neural analysis**, enabling biologically grounded interpretability of deep RL agents.
(3) Using these tools, we show that model-free RNN agents exhibit structured, planning-like behavior through emergent dynamics, despite lacking explicit world models.
(4) We release a reproducible pipeline and open-source codebase to support further research on internal representations and behavior.
We demonstrate these contributions in ForageWorld, analyzing trained agents across **five motif classes: exploration, decision-making, learning dynamics, internal representations, and generalization.**

The remainder of the paper reviews related work (Section 2), introduces our task and analyses (Section 3), and presents results (Section 4).

Table 1: *Understanding DRL agent behavior in open-ended tasks requires context-sensitive analysis frameworks, much like those used in animal neuroscience.* This table summarizes our reusable toolkit for probing memory, planning, exploration, and internal representations in DRL agents.

| Analysis Target | Key Question | Methods Used |
| --- | --- | --- |
| Goal Inference | What goals is the agent pursuing, and how are they shaped by experience? | Decision GLMs, in-context learning study, behavior phase segmentation |
| Memory Span | How far back in time can the agent remember? | RNN state decoding, memory module ablations |
| Planning Horizon | How far ahead does the agent plan? | Future decoding, auxiliary loss analysis for predictive objectives |
| Spatial Structure | How does the agent move through the environment? | Position occupancy entropy, tracking revisitation, path analysis |
| Network Capacity | Is the model too large or too small for the task difficulty? | Network size/pruning sweeps, comparing performance curves |
| Representations | What information is encoded in the agent's internal states? | Decoding, GLMs, encoding profiles of task variables |

## 2   Background: Open-Ended RL as a Bridge to NeuroAI

### 2.1   Open-Ended RL and the Need for Behavioral Analysis

Generalization in RL refers to the ability to transfer knowledge to new environments or tasks, and is critical for both practical applications and theoretical understanding in AI [26]. However, many RL algorithms struggle to generalize beyond their training tasks [27, 28]. This bottleneck has renewed interest in open-ended RL tasks that require rich exploration, subgoal discovery, and long-horizon reasoning [1, 29]. Open-ended RL tasks typically (i) scale with the learner, requiring skill composition in large state spaces, and (ii) incorporate naturalistic structure to better reflect real-world environments [29]. Many recent benchmarks embody this open-ended perspective—emphasizing compositional skills, dynamic goals, procedural generation, and expansive state spaces[1, 11–13, 30–39].

### 2.2   Joint Behavioral and Neural Analyses for Artificial Agents

To date, open-ended RL benchmarks have not drawn sustained interest from neuroscience or cognitive science, leading to a lack of deeper analysis frameworks in this research area. One reason is that few open-ended tasks reflect how animals explore, plan, and decide [3, 40]. Thus, there is a need for more naturalistic benchmarks—and for analysis pipelines that study agents with the same rigor applied to animal behavior and neural computation. Recent neuroscience and cognitive science work has primarily applied brain-inspired tools to ML systems for comparing internal representations [41–50]. This lack of behavioral emphasis mirrors limitations in how internal dynamics are typically studied—often in isolation from behavior, or without the tools needed to reveal structure. To date, most of this work has focused on representational similarity, not on joint analysis of behavior and underlying neural computation. A few recent exceptions—mostly outside RL—have begun to explore interpretability with a greater focus on behavior [47, 51, 52, 13, 53]. Researchers have also made calls for more interpretability work in the RL domain as well [54–56].

Despite this challenge, a key lesson from neuroscience is that neural activity alone is often insufficient to explain computation [3, 57–60]. For example, recent findings show that common similarity metrics can yield inconsistent conclusions [48, 61]. Several recent works argue that neuroAI should prioritize aligning behavior and internal dynamics in naturalistic settings, rather than relying on coarse similarity metrics in toy tasks [62, 60]. We thus adopt a behavior-first approach to studying DRL agents in the naturalistic, open-ended task ForageWorld, characterizing their strategies and internal representations using neuroscience-inspired tools—as is done in biological systems.

## 3   Methods

This section summarizes the environment, model architectures, training procedures, and analysis framework we used in this work. Full code for ForageWorld and our analysis is available at `https://github.com/RileySE/Craftax-Foraging/tree/foraging`, with additional details in the appendix.

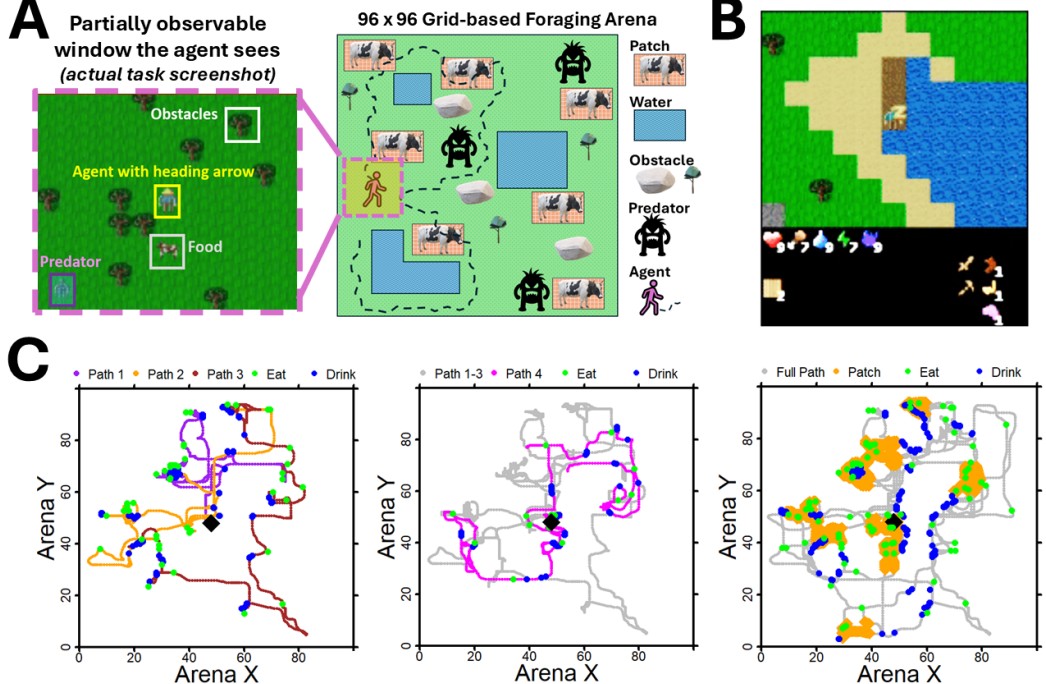

Figure 1: *Model-free agents exhibit structured exploration and revisitation behavior in an open-ended, partially observable environment.* (A) A 9×11 grid view (left, local observation window) shows the agent's local observation, positioned within the full 96×96 environment (right). Cows (food) diffuse from spawn points and deplete when consumed; lakes (drink) are fixed and unlimited; predators intermittently pursue agents in view. (B) Agent observations include the visual window, inventory, and internal states (e.g., health, hunger, fatigue). This agent is sleeping—typically in corners to reduce predator contact. (C) Trajectories from a single episode: three early sequential exploratory paths (left), initial revisitation (middle), and full path with revisited patches in orange (right). The paths depicted belong to the same episode, and are directly consecutive from Path 1 through 4 in the behavioral time series logs for the episode. Path 1 starts at t=0 of the episode.

## 3.1 ForageWorld: Naturalistic Foraging with Internal States

We build on the Craftax benchmark [12], modifying it to support more naturalistic behavior under partial observability. ForageWorld introduces ecological constraints and uncertainty to better reflect real-world cognitive demands and enable deeper analysis. In particular, the need to localize food and avoid predators increases the memory demands relative to the original Craftax task. Cows have an arena-wide count limit, spawn at fixed locations, and diffuse over time—leading to temporary, localized food depletion and requiring the agent to leave and revisit patches periodically (Figure 1).

Drinking from procedurally placed water sources maintains hydration. Predators intermittently appear near food patches and pursue agents when within line of sight (Figure 1). Agents manage hunger, thirst, fatigue, and health. Fatigue recovers only through sleep, starvation and extreme thirst reduce health, and death occurs if the health variable hits zero. Sleep can only be performed when energy is below 50% of maximum. It immobilizes the agent while gradually restoring energy and, if applicable, health (Figure 1). Arenas are procedurally generated with randomized layouts, ensuring per-episode novelty. Unlike Craftax, our reward function is focused on agent survival: to receive reward, the agent must maintain its physiological resources (health, food, drink, energy) above half their maximum values. This objective correlates strongly with survival time and encourages the agent to maintain a buffer in resource levels—without promoting overconsumption or maxing out all variables. Specifically, the reward function is:

$$\mathcal{R}(s_t) = 0.1 \times \big(1 + \text{sign}(\text{health}_t - 5) + \text{sign}(\text{food}_t - 5) \\ + \text{sign}(\text{drink}_t - 5) + \text{sign}(\text{energy}_t - 5)\big) \quad (1)$$

As defined in Equation 1, the agent's reward is a function of physiological variables: it receives increasing positive reward for each key survival resource maintained above a threshold—health, food, drink, and energy.

For analysis, we log neural states, actions, and environment variables for each timestep (Table 2). This setup also supports held-out evaluation, enabling generalization assessment. Logged variables span both behavioral and neural axes from our analysis framework (Table 1).

## 3.2 RL Architectures and Model Variants

Our primary architecture was PPO with a GRU-based recurrent core (PPO-RNN), detailed in subsection A.6. In some experiments, we added an auxiliary objective to encourage interpretable spatial representations. In these experiments, the model also outputs its predicted position $(x_t, y_t)$ relative to the first-timestep origin (path integration), via a small fully connected head that shares the RNN hidden state $h_t$ with the policy and value heads. We train this auxiliary objective jointly with the PPO loss, using an L2 regression loss:

$$L_{\text{aux}} = \mathbb{E}_t[\|\hat{p}_t - p_t\|_2^2] \tag{2}$$

This term is weighted by a factor $\mathcal{W}_{\text{aux}}$ during optimization. This objective encourages RNN states to encode location, enabling spatial decoding (e.g., allocentric structure, planning horizon) and downstream analysis. Examining internal dynamics allows us to link behavior to emergent representations—paralleling methods in animal navigation studies.

To introduce biologically-inspired sparsity, we prune the weights of the RNN core using `JaxPruner` during training [63]. We targeted 90% sparsity across model layers, approximating biological brain sparsity, which ranges from 70% to 94% across species [64, 65]. We used magnitude-based pruning. Pruned agents performed comparably to unpruned ones and, in some cases, showed improved decoding of spatial variables from hidden states (Figure 16)–consistent with findings that sparse connectivity improves modularity and interpretability [66–68].

## 3.3 Decoding Allocentric Position and Planning Horizons

To evaluate memory and planning, we test whether hidden states encode past and future spatial positions. If location is internally represented, it may enable revisitation and planning. We first record hidden states $h_t$ and ground-truth positions $(x_t, y_t)$ throughout each episode. Then, at each timestep $t$, we train a decoder to predict allocentric displacement $Y_{t+\Delta t} = (\Delta x, \Delta y)$ between the current hidden state $h_t$ and the agent's position $\Delta t$ steps in the past or future, using only $h_t \in \mathbb{R}^{512}$. This method tests whether the RNN encodes allocentric spatial information at various planning and memory horizons—analogous to representations in animal navigation systems. Additionally, we train separate decoders for allocentric (relative to origin) and egocentric (relative to body orientation) coordinates to assess which coordinate frame the agent uses internally. We use ridge regression as the decoding model to preserve interpretability. Decoders were trained on the first 75% of each episode and evaluated on the final 25% to assess generalization. Accuracy is reported as RMSE, benchmarked against an average displacement-based baseline. All behavioral analyses were conducted in held-out test arenas, with weights frozen throughout, to ensure that behavior reflects generalization rather than ongoing learning. See subsection C.2 and subsection C.3 for full training details.

# 4 Results

## 4.1 Structured Exploration and Revisitation in Novel Environments

We begin by examining how agents explore novel environments and transition to revisitation strategies. Foraging has long served as a rich behavioral paradigm for studying goal-directed exploration in animal neuroscience [69, 70]. In rodents, foraging in novel, partially observable environments unfolds in distinct behavioral phases: mice initially engage in broad exploration, followed by direct and efficient revisitation of previously discovered locations once the environment is mapped [71].

Using trajectory visualizations, behavioral segmentation, and spatial entropy metrics, we find that ForageWorld induces structured, phase-like foraging behaviors in trained DRL agents—analogous to those observed in biological systems. Standard performance metrics confirm that agents are competent at the task, showing long survival times (directly proportional to return) after training (Figure 2). However, our behavioral analysis reveals that DRL agents equipped only with RNNs—i.e., without explicit world models—develop in-context adaptive learning and exploration strategies that evolve across episodes.

Strikingly, after initially mapping a novel arena, agents shift into revisitation behaviors that reflect structured, multi-objective decision-making—mirroring phase transitions in animal foraging [71] and information seeking [72–74]. Their early exploration trajectories qualitatively resemble known

patterns of insect search behavior (Figure 8). Agents generated outwardly spiraling, azimuthally rotating loops from their starting position that gradually expanded to cover most of the arena (Figure 1). Even during revisitation phases, exploration persisted—most successful agents had covered the majority of the arena by the end of each episode (Figure 6). These extended trajectories also supported predator avoidance, enabling agents to discover shortcut paths between safe regions and resource patches (Figure 7). Similar patterns have been observed in ants and bees during nest displacement or early flights from the hive (Figure 8).

## 4.2   Multi-Objective Foraging and Strategic Patch Use

Having characterized agents' exploration and revisitation behaviors, we next examine how they select among remembered locations. After initial exploration, agents displayed a clear shift to revisitation—returning to remembered patches via direct routes. Similar to rodents navigating partially observable mazes [71], our agents exhibit a rapid transition from broad exploration to targeted revisitation (Figure 1). This transition mimics phase-structured foraging as seen in rodents, while exploration resembles insect search behavior (Figure 8). Agents also periodically return to their starting position—potentially a reorientation strategy in the absence of an explicit spatial map.

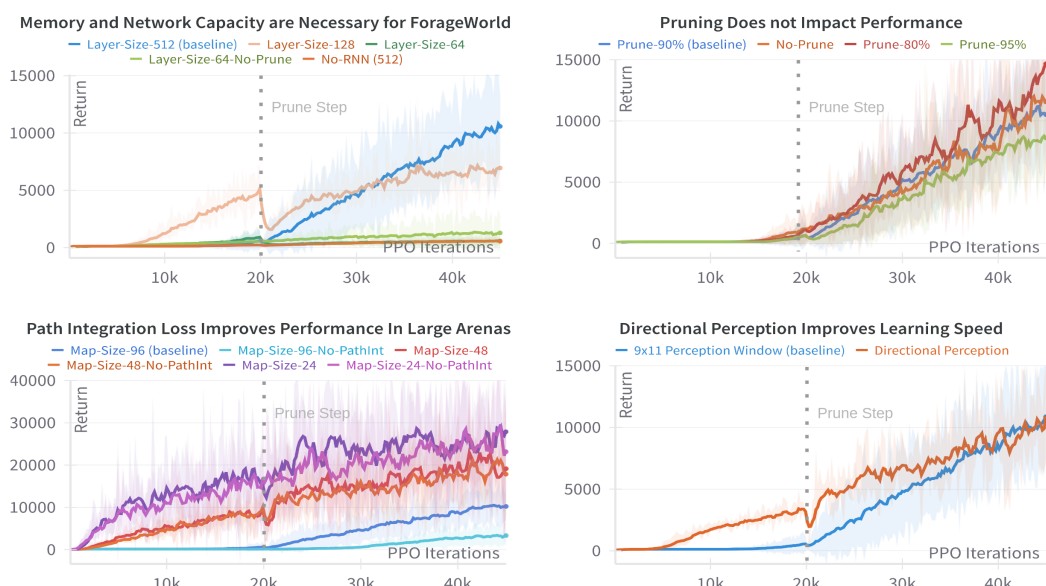

Figure 2: *Performance metrics and model ablations motivate deeper behavioral-neural analysis.* (**Top left**) ForageWorld requires memory and substantial network capacity to learn. Replacing the 512-unit RNN with a feedforward network significantly impairs performance, as does downsizing to a 64-unit network (550k parameters). A midsize 128-unit network also underperforms when pruned, suggesting that high capacity is needed to support sparsity. (**Top right**) Pruning does not degrade training performance but improves the spatial interpretability of internal representations (see Figure 16). (**Bottom left**) Removing the auxiliary path integration loss reduces performance in large arenas—but not small ones. Thus, the ability to predict current self-position appears critical for larger-scale navigation (see Figure 17 for representational effects). (**Bottom right**) Limiting perception to a forward-facing field of view improves early learning—perhaps due to reduced input complexity—but final performance conceals behavioral differences (see Figure 15). (**All plots**) Curves show the time-weighted EMA mean across 5 random seeds; shaded regions show one standard deviation.

Agents are not given an explicit list of patch locations. Instead, revisitation behavior emerges from recurrent state dynamics encoded during exploration and is reconstructed for analysis from logged trajectories. This result shows that model-free RNN agents can develop memory-guided, goal-directed behavior—such as selective revisitation—without explicit world models or symbolic memory structures. This memory dependence is further supported by a substantial performance gap between PPO agents with and without recurrence (Figure 2, Figure 11, Figure 10). Like animals that balance intake against physiological constraints (e.g., stomach volume, satiation), our agents are rewarded for maintaining low hunger, thirst, and fatigue—rather than for rapidly depleting resource patches. Under our baseline task parameters, expert agents ate at an average rate of 0.011 (approximately once

every 95 steps), drank at 0.034, and slept at 0.262. This emergent spacing enabled us to test whether agents made memory-guided, multi-objective decisions when selecting which patches to revisit. We analyzed patch revisitation choices and found that agents integrated spatial and task-relevant features in a multi-objective manner (Figure 3, Figure 7).

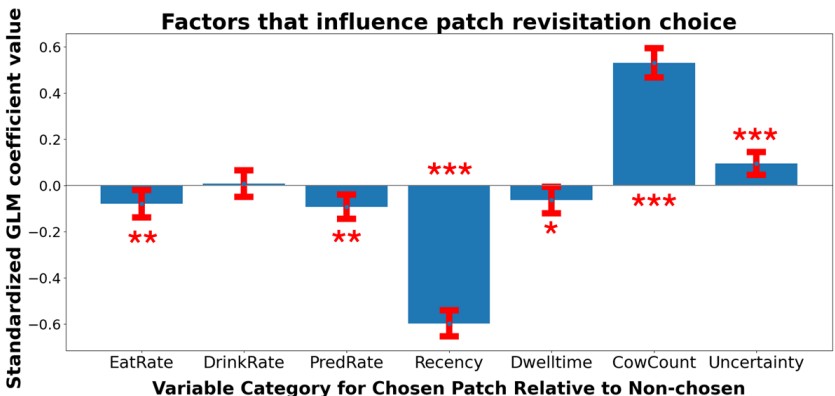

Figure 3: *Memory-guided, multi-objective revisitation strategies emerge in model-free agents without world models.* Generalized linear model (GLM) coefficients for patch history variables predicting an agent's choice to revisit one patch over others. Choices are defined 50 timesteps before each patch eat event. From left to right: agents prefer patches with fewer prior eat actions (EatRate); show no preference for water proximity (DrinkRate); avoid patches with more predator encounters (PredRate); prefer patches visited more recently (Recency); show mild preference for longer dwell time (Dwelltime); prefer patches with more observed cows (CowCount); and prefer patches with higher prior position prediction error (Uncertainty). Significance levels: $* < 0.05$, $** < 0.01$, $*** < 0.001$. Figure 13 has model output and VIF analysis. Error bars are 95% confidence intervals (CI).

### 4.3 Emergence of Behavioral Competencies Over Training

To examine how agent behavior evolves during training, we analyzed performance metrics and behavioral patterns across episodes. We find that ForageWorld supports staged learning: agents shift from undirected exploration to structured, goal-aligned strategies—transitions only partially visible in standard reward curves. These patterns suggest temporally organized skill acquisition. To test this, we tracked how different behaviors emerged and stabilized over the course of training. We used a panel of behavioral metrics to characterize the timeline of skill acquisition and identify which behaviors drove early versus late performance gains (Figure 4, Figure 9).

Early in training, PPO-RNN agents exhibited a "fishing" strategy—remaining stationary near the origin for extended periods. After approximately 20,000 training iterations, agents underwent an abrupt shift in behavior across multiple metrics. This transition included longer-distance travel during both early exploration and later revisitation, more strategic trade-offs between water and food collection, a sharp increase in tool-making frequency, and the emergence of gradual gains in predator defense. This shift marks the emergence of higher-level strategies in the training process.

To evaluate the contribution of specific components of the environment and model, we compared our PPO-RNN baseline to several variants with targeted objectives or capacities removed. One key ablation altered the agent's field of view (FOV) to include only the forward grid rows, rather than the default Craftax FOV of a 9×11 rectangle centered on the agent. This change increased the need for information-seeking behavior and more closely resembled animal vision, which is predominantly front-facing. We found that front-FOV agents explored farther earlier in training and often engaged in more localized, dense search patterns (Figure 14, Figure 15). We also trained a different DRL objective, an off-policy RL architecture called parallelized Q-network (PQN) [75] (Appendix E). PQN-LSTM models performed comparably to the PPO-GRU baseline Figure 29, and with overall similar learning histories and pathing (Figure 31, Figure 30), but with the noticeable behavioral differences of much lower rates of predator killing (i.e. reducing predator HP to 0) (Figure 31). Thus, PPO-GRU seems to converge to a predator fighting strategy, while PQN-LSTM converges to predator evasion.

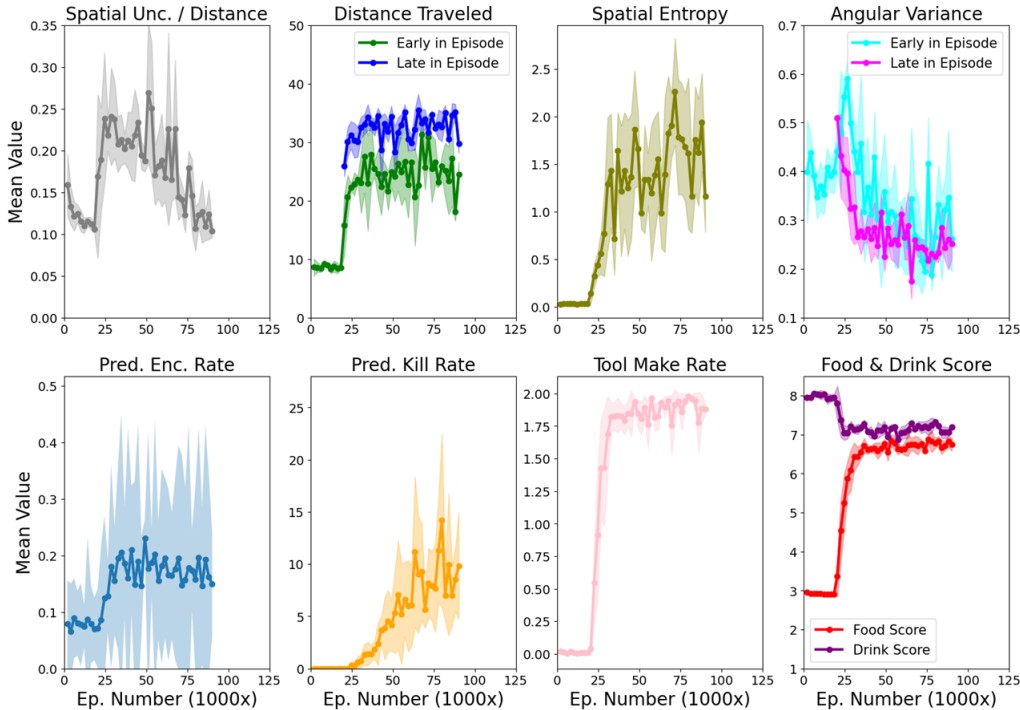

Figure 4: *Distinct behavioral competencies emerge over training, with early gains in exploration followed by refinement of survival strategies.* Training dynamics show staged acquisition of task-aligned skills. Early learning emphasizes spatial exploration and arena coverage, while later training refines predator response, tool use, and pathing patterns. Metrics include: spatial uncertainty (normalized by distance from origin); distance from origin (early vs. late, meaning before vs. after the first 1500 timesteps in an episode); state occupancy entropy of agent position; angular orientation variance across 250 timestep intervals; predator field-of-view exposure; tool-making rate (1 = one tool crafted, 2 = both); and food/water satiation levels. Error bars are 95% CI. These plots can be compared to the generally linear performance gains seen in survival times across training in Figure 2.

## 4.4 Recurrent State Representations Support Memory and Planning

We next ask whether agents encode task-relevant structure in their internal representations. Using linear decoders trained on recurrent states, we find that agents represent allocentric position across time—revealing implicit memory and planning capacity not apparent from performance metrics alone. Thus, our framework can (1) decode the memory and planning timescales represented in agent RNN states, and (2) identify the internal encoding structure that supports foraging behavior.

A growing body of work in RL has examined whether model-free RNN agents can perform implicit planning without the explicit world models typically used for planning tasks [76–78]. This question also arises in neuroscience, where insects like ants and bees show planning-like behavior despite lacking mammalian hippocampal-entorhinal map systems [79–81]. Our analyses show that DRL agents revisit patches based on multiple factors such as proximity, predator history, and position prediction error (Figure 3, Figure 7). An agent's position relative to the origin can also be decoded from its current RNN state—up to 50–100 timesteps into the past and future—supporting its capacity for long-horizon memory and planning (Figure 5). Furthermore, approximately 100/512 neurons were found to be position-sensitive at the single neuron level using generalized linear model (GLM) analysis. The neural activity response to position change (GLM coefficient magnitude) increased with radial distance from the agent's origin as well, suggesting a distance accumulation circuit that modulates origin-revisitation behavior (Appendix D).

Decoding accuracy improved over the course of training, consistent with the agent acquiring more stable and structured internal representations. In contrast, the agent's position relative to itself (egocentric orientation) could not be reliably decoded. This preference for origin-relative encoding may help explain why agents frequently revisit the arena origin during later phases (Figure 1).

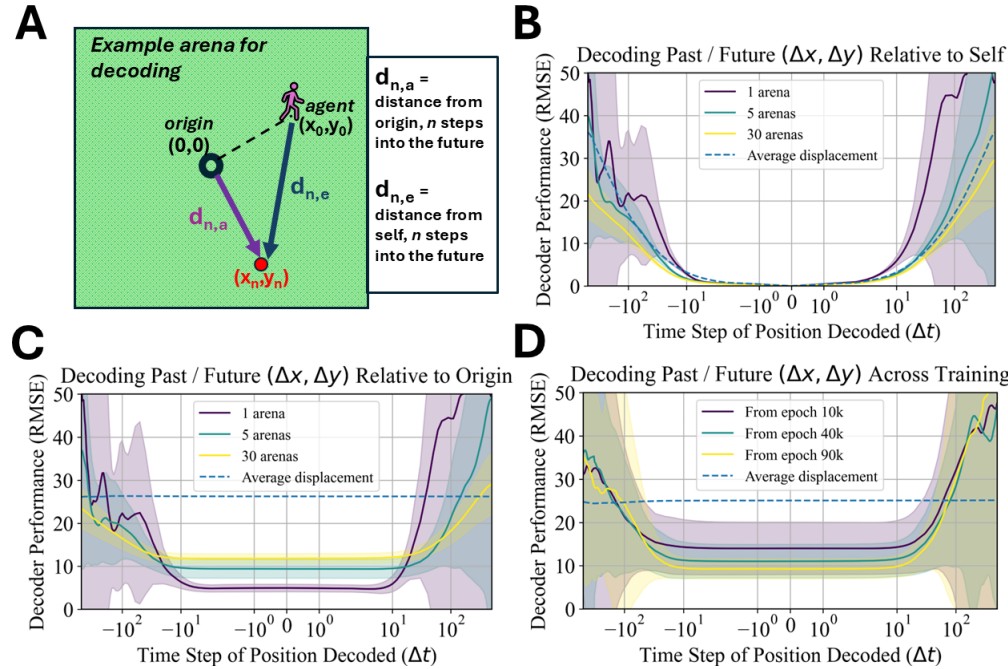

Figure 5: *RNN states encode allocentric position and temporal structure, revealing emergent memory and planning capacity in model-free agents.* (A) A single decoding model was trained per agent to test whether spatial information was encoded in allocentric (relative to origin) or egocentric (relative to agent) coordinates. (B) Late-training decoding performance for egocentric distance remained at chance level across models. (C) Allocentric position could be decoded above chance up to approximately 50–100 timesteps into the past and future depending on the run. Training arena count per decoder was varied to show that models did not rely on arena-specific cues. (D) Allocentric decoding improved over training. Error bars are larger in (D) because decoding was limited to timesteps 1000–6000 per arena, due to shorter survival in early-training (10k epoch) agents. All plots use average displacement per timestep as a chance baseline. Error bars reflect 95% CI.

## 4.5    Generalization and Modularity in Recurrent Dynamics

We next examine whether the internal representations learned by DRL agents generalize across environments and exhibit modular substructure. These properties are important not only for performance, but also for interpretability—revealing which units encode what, and under what conditions. Decoding performance was moderately higher in pruned (sparse) networks than in fully connected ones (Figure 16), confirming more efficient information encoding in sparse architectures—consistent with recent findings [66–68]. Removing the auxiliary objective for position prediction eliminated above-chance decoding performance (Figure 17), suggesting that the associated task performance loss (Figure 2) stems from the model's inability to encode position information using the PPO objective alone. These results suggest that the auxiliary objective improved both agent performance and the interpretability of its internal representations—a dual benefit revealed only through neural analysis. Past and future position encoding relied on overlapping but functionally modular subpopulations of RNN units, which diverged more clearly at longer time horizons (Figure 18). Notably, a single decoding model generalizes across episodes—predicting past and future position from RNN states recorded in many different arenas for the same agent. This cross-episode generalization likely reflects the consistent spatial structure of the environment: although arena configurations vary, the 96×96 grid layout and global orientation are preserved, providing a stable allocentric reference frame.

In summary, model-free RNN agents trained in open-ended tasks can develop structured internal representations that support memory, planning, and spatial generalization—without relying on explicit world models. ForageWorld induces naturalistic behavioral and neural motifs in DRL agents, and our analysis framework enables these motifs to be detected and interpreted using behavior-first tools.

# 5 Discussion

We provide two main findings: (1) ForageWorld induces behavior–neural motifs relevant to planning and memory, and (2) our toolkit reveals these motifs in ways that simple performance metrics cannot.

We also help challenge the common view in neuroscience that sophisticated planning and memory require explicit world models, mammalian-like brains, or symbolic memory [80, 82], a topic that is of growing interest in computer science as well [83, 84]. Despite being trained with model-free objectives, RNN-based agents in our study exhibit revisitation, uncertainty-driven exploration, and allocentric encoding—behaviors typically associated with model-based agents. While recent work suggests that temporal credit assignment and environment interaction can give rise to planning-like behavior in RNNs [76–78], our study provides the first systematic evidence that such behavior can emerge in complex, naturalistic environments without world models. Thus, internal interpretability is shaped not just by training outcomes, but by design choices that constrain or promote structure.

We also find that architectural choices such as recurrence, pruning, and auxiliary losses significantly influence both performance and interpretability. Pruning improved decoding accuracy (Figure 16) while leaving performance and behavior substantively unchanged (Figure 2); the auxiliary position prediction objective enhanced predator-related behavior and spatial encoding structure (Figure 4, Figure 11, Figure 17) while also improving performance (Figure 2). This win-win fits with observations from neuroscience—biological agents do not simply maximize reward, but also engage in latent path learning and other subgoals [82]. Thus, there are benefits for RL researchers to embracing this complexity of learning objectives and bridging the gaps between artificial and natural intelligence.

These findings also speak to neuroscience research on small nervous systems. Insects, for example, navigate using structured internal cues [80, 85]. The fact that similar behaviors emerge in DRL agents with only a few hundred recurrent units suggests that such agents may offer useful models for evaluating the cognitive capacities of small nervous systems. Taken together with behavioral consistency across arenas, the ability of a single decoding model to generalize across episodes likely reflects a stable, compass-like encoding of space—mirroring strategies used by insects and other animals to maintain orientation using global cues such as magnetic fields or polarized light [86–88]. This encoding strategy may explain how DRL agents reliably plan navigation across environments with differing local layouts.

More broadly, we dispute the idea that DRL models are too complex for neuroscience-style analysis. The brain is at least as complex—and likely more so—than current DRL agents [45]. If neuroscience tools cannot explain DRL agents with full behavioral and neural access, their utility for understanding the brain is limited. Testing these tools in complex simulation environments also addresses practical challenges in naturalistic neuroscience, including low statistical power, high data costs, and limited experimental control [89]. By generating high-volume, ecologically grounded datasets with joint neural-behavior logging, we provide a platform for evaluating brain analysis techniques. Our diagnostic framework (Table 1) offers a guide for interpreting agent behavior and internal dynamics across diverse RL tasks. We hope this framework serves both as a roadmap for future neuroAI work and as a foundation for more interpretable RL evaluation.

**Limitations & Broader Impacts** The $96 \times 96$ grid size limit in Craftax prevents us from evaluating how our results generalize to even larger arenas. A second limitation is that we could not log the precise locations of all trees, rocks, and lakes, which limited our ability to study landmark-based navigation in detail. A third limitation is that incorporating a wider range of RL architectures with full behavioral-neural logging is currently non-trivial in this environment, which is why we focused on PPO- and PQN-based models with shared components. We believe this work is likely to have a positive societal impact by fostering closer collaboration between neuroscientists, cognitive scientists, ethologists, and RL researchers in the analysis of deep RL agents—advancing all fields through shared insight into the structure and function of learned behavior.

# 6 Acknowledgments

We thank Albert Lee and members of the Rajan Lab for the helpful comments and feedback during the course of the project. We also thank the research assistants Yang Wu and Yunxin Wu for their help during revisions. R.S., R.P.B. and K.R. were collectively funded by the NIH (RF1DA056403, U01NS136507), James S. McDonnell Foundation (220020466), Simons Foundation (Pilot Extension-00003332-02, McKnight Endowment Fund, CIFAR Azrieli Global Scholar Program, NSF (2046583), Harvard Medical School Neurobiology Lefler Small Grant Award, Harvard Medical School Dean's Innovation Award, and Harvard Medical School Leonard and Isabelle Goldenson Fellowship. This work has been made possible in part by a gift from the Chan Zuckerberg Initiative Foundation to establish the Kempner Institute for the Study of Natural and Artificial Intelligence at Harvard University. F.B.B. received support from Fulbright Program, Jansons Legat, and the Norwegian State Educational Loan Fund.

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

# 7 Appendix

This Appendix provides additional details on the ForageWorld environment and agent training setup, supplementing the descriptions in subsection 3.1 and subsection 3.2 of the main text. It includes key environment parameters, procedural generation constraints, agent sensory input, and logging infrastructure used for downstream analysis. We also provide supplementary analyses and figures that expand on the main results, including trajectory examples, revisitation models, training dynamics, ablations, and neural decoding structure.

## A ForageWorld: Task and Training Details

We designed ForageWorld to balance between task complexity, transparency to analysis, training efficiency (using JAX GPU acceleration), biological plausibility in navigational planning computations, and connection to joint RL and recurrent memory architectures that are of particular interest to neuroscientists. More complex deep RL tasks exist, but many such tasks (for example, Minecraft) are much more difficult to study and in particular more difficult to instrument for behavior analysis. In addition, the increased computational cost of experiments in such environments is prohibitive to some statistical analysis, and many computational neuroscientists in particular do not have access to compute resources sufficient to run experiments in such complex environments. Relative to common neuroscience tasks used to study foraging, e.g. n-armed bandit tasks which lack spatial or temporal components, ForageWorld pushes the complexity boundary on the neuro-side of neuroAI while still remaining tractable for neuroscience tools and neuroscientists to study. Further, ForageWorld as it exists is not trivial to learn— our agents require billions of timesteps of experience to train to competency on the task, and Craftax, the benchmark which we build upon to develop ForageWorld, has not been solved by any agent or architecture to date.

In the following section, we expand here on the ForageWorld environment and agent training setup described in subsection 3.1 and subsection 3.2 of the main text. This includes key environment parameters, procedural generation constraints, agent sensory input, and logging infrastructure used for downstream analysis.

### A.1 Procedural Arena Generation

Each arena is procedurally generated at the start of an episode using a fixed-size $96 \times 96$ grid. Spawn points for cows (food) and lakes (drink) are initialized in random patches using Perlin noise, providing episodic variability alongside naturalistic structure. Obstacles are added to break line-of-sight and create navigational constraints. The agent begins at the center of the arena.

### A.2 Episode Structure and Termination Conditions

Each episode runs for a maximum of 100,000 timesteps or ends earlier if the agent's health drops to zero—which is the typical outcome, even for highly competent agents. Agents earn reward by maintaining internal homeostasis—balancing hunger, thirst, and fatigue—and avoiding predator encounters. The reward function is defined in subsection 3.1.

### A.3 Sensory Input and State Representation

At each timestep, the agent receives a $9 \times 11$ grid-centered egocentric view of the environment (see Figure 1), along with an inventory vector containing current health, satiation levels (food and water), fatigue, and any collected items. This full observation is encoded by a feedforward neural net layer before being passed to the recurrent network.

### A.4 Logging Infrastructure

To support joint behavioral and neural analysis, we log a wide range of signals at each timestep, including the full recurrent hidden state, internal reward components, movement trajectories, environmental state, and agent decisions. The complete logging schema is shown in Table 2. These logs enable replay, ablation, and representational analysis across multiple analytic pipelines.

## A.5 Training Hyperparameters

Agents were trained using Proximal Policy Optimization (PPO) [90] with Generalized Advantage Estimation (GAE). Each agent trained for 3 billion timesteps, resulting in a variable number of episodes—typically in the hundreds of thousands—depending on competence and learning speed. Episodes were capped at 100,000 timesteps, though this cap was rarely reached in practice. The PPO rollout horizon was 64, with a minibatch size of 8192 and a learning rate of 0.00025. We used the Adam optimizer with default momentum parameters. Additional hyperparameters are provided in Table 3. Policy and value losses were weighted using standard PPO coefficients (see Table 3). For stability, gradient norms were clipped to 1.0.

Each training run used a single GPU (NVIDIA A100 or H100) and typically converged within 24–48 hours, depending on configuration. While memory usage varied by configuration, most experiments were feasible on GPUs with at least 24 GB of memory. Logging occurred every 2048 PPO iterations (roughly every 134 million timesteps), and the best-performing checkpoints were selected based on held-out arena performance.

Together, these settings define a procedurally rich, ecologically grounded RL training setup that supports the emergence of complex behaviors and structured internal representations. The design balances task realism, computational feasibility, and interpretability—enabling the behavioral and neural analyses presented in this work.

## A.6 Model Architecture Details for PPO-RNN

Our primary architecture combined Proximal Policy Optimization (PPO) [90] with a recurrent neural network, using Generalized Advantage Estimation (GAE) for stability. PPO optimized a clipped surrogate objective that balanced advantage maximization with policy stability:

$$L_{\text{clip}}(\theta) \ = \ \hat{\mathbb{E}}_t \Big[ \min\Big( r_t(\theta)\,\hat{A}_t, \ \text{clip}\big(r_t(\theta),\, 1-\epsilon,\, 1+\epsilon\big)\,\hat{A}_t \Big)\Big], \tag{3}$$

where $r_t(\theta) = \frac{\pi_\theta(a_t|s_t)}{\pi_{\theta_{\text{old}}}(a_t|s_t)}$ is the probability ratio between current and previous policies, and $\hat{A}_t$ is the GAE-computed advantage. The full PPO loss also includes value and entropy terms, following standard practice.

To handle partial observability in the task, the agent uses a gated recurrent unit (GRU) [91] to build memory over time. At each timestep $t$, the GRU took the current observation $o_t$ and previous hidden state $h_{t-1}$, producing an updated hidden state $h_t$. Both the policy $\pi(a_t \mid o_{\leq t})$ and the value estimate $V(o_{\leq t})$ are computed from $h_t$ via separate output heads. This architecture is shared between the actor and critic networks. We used a single-layer GRU with 512 hidden units. GRUs offer a gating mechanism that supports long-range memory retention while being more computationally efficient than LSTMs. This advantage makes them especially well suited for tasks involving memory and interpretability. We followed the implementation provided in the Craftax benchmark [12], which demonstrated that memory-equipped PPO agents perform substantially better on partially observed, long-horizon tasks. In total, our standard configuration contains approximately 6.5 million trainable parameters.

## A.7 Future directions in ForageWorld

The Craftax platform also provides an opportunity for future work not yet explored here: incorporating curriculum learning into ForageWorld and related environments [92, 93]. The staged emergence of behavioral competencies we observed (e.g., in Figure 4) suggests that ForageWorld naturally supports curriculum-based training regimes and Craftax is compatible with JAX curriculum learning packages. We hope future work will explore curriculum learning [92, 93]. Future studies could leverage this structure—and the motif-based analysis framework—to probe how learning history shapes internal representations, and how staged training affects generalization, revisitation strategies, and neural dynamics. Along this direction, it would also be interesting to test LLM models' foraging behavior, as navigation is an underexplored topic in LLM literature generally [94] (though note that transformers and LLMs would be harder to analyze than the simpler RNNs chosen here for their brain-like recurrent activity). However, Craftax is currently not coded for language-based labels of state variables so additional modification would have to be done to support LLM tests.

Lastly, note that Craftax and ForageWorld do not support multi-agent environments yet, but this would be an interesting modification to explore.

# B    Interpreting Agent Behavior: Supplementary Figures and Analysis

This section provides supplementary behavioral analyses and supporting visualizations. It expands on the main text results in section 4, including example trajectories, phase structure segmentation, and multi-objective revisitation strategies. We also present ablations targeting memory, auxiliary objectives, and predator combat to support interpretability and robustness claims.

## B.1    Structured Exploration and Revisitation Trajectories

We present representative trajectories from trained agents that illustrate the structured exploration and patch revisitation behaviors described in section 4. Figure 6 shows trajectories from three episodes, with early exploratory loops followed by increasingly direct revisitation as the environment becomes familiar. These examples demonstrate that agents undergo a consistent shift from broad exploration to memory-guided foraging, even in novel arenas.

## B.2    Behavioral Phase Segmentation via Unsupervised Clustering

To understand how agents modulate their movement patterns across time and task demands, we applied unsupervised clustering to the agent's locomotion features. We used the `bayesmove` package to identify three latent movement states based on turning angle and step size over a 7-timestep moving window [95]. We next used conditional inference trees to identify which task variables best predicted transitions between these states [96].

As shown in Figure 7, the identified states correspond to interpretable movement modes: short-range (state 3), mid-range (state 1), and long-range (state 2) navigation. These were differentially associated with predator presence, eating behavior, and positional uncertainty. In particular, state 1 frequently coincided with predator events, suggesting an evasive or scanning behavior, while state 3 tended to occur near food and was enriched for eat actions. This analysis highlights the agent's ability to adapt its movement patterns in response to internal and external signals, without requiring explicit mode-switching logic.

## B.3    Comparison to Real Insect Search Patterns

To contextualize these emergent strategies, we compared agent trajectories to documented search behaviors in ants and bees. As shown in Figure 8, both species exhibit progressively expanding search loops when navigating novel or ambiguous environments. These loops are typically centered around a known location, such as a hive or nest. These patterns strongly resemble the outward-spiraling, azimuthally rotating loops exhibited by our agents during early exploration in novel ForageWorld arenas. Although the sensory and neural mechanisms differ across biological and artificial systems, the structural similarity in their trajectories suggests that simple control policies—when shaped by partial observability, uncertainty, and survival constraints—may converge on similar search dynamics.

## B.4    GLM Outputs and Multicollinearity Checks

To support our analysis of patch revisitation strategies, we fitted generalized linear models (GLMs) to evaluate the influence of task variables on agent decision-making. Each GLM was trained to classify whether a patch was chosen at a revisitation point, using historical variables—such as recent reward rates, predator presence, and spatial uncertainty—as input features. Each model was trained on data from multiple agents and episodes, with agent identity included as a fixed effect to account for inter-agent variability.

Figure 13 shows the full GLM outputs underlying Figure 3, as well as variance inflation factor (VIF) scores used to assess multicollinearity among the regressors. As expected, uncertainty, predator encounter rate, and recency of visitation emerged as significant predictors, while variables such as dwell time and drink rate contributed less. VIF analysis confirmed that none of the regressors exhibited problematic multicollinearity. While revisitation behavior captures spatial strategy, we also tracked how decision confidence evolved over training.

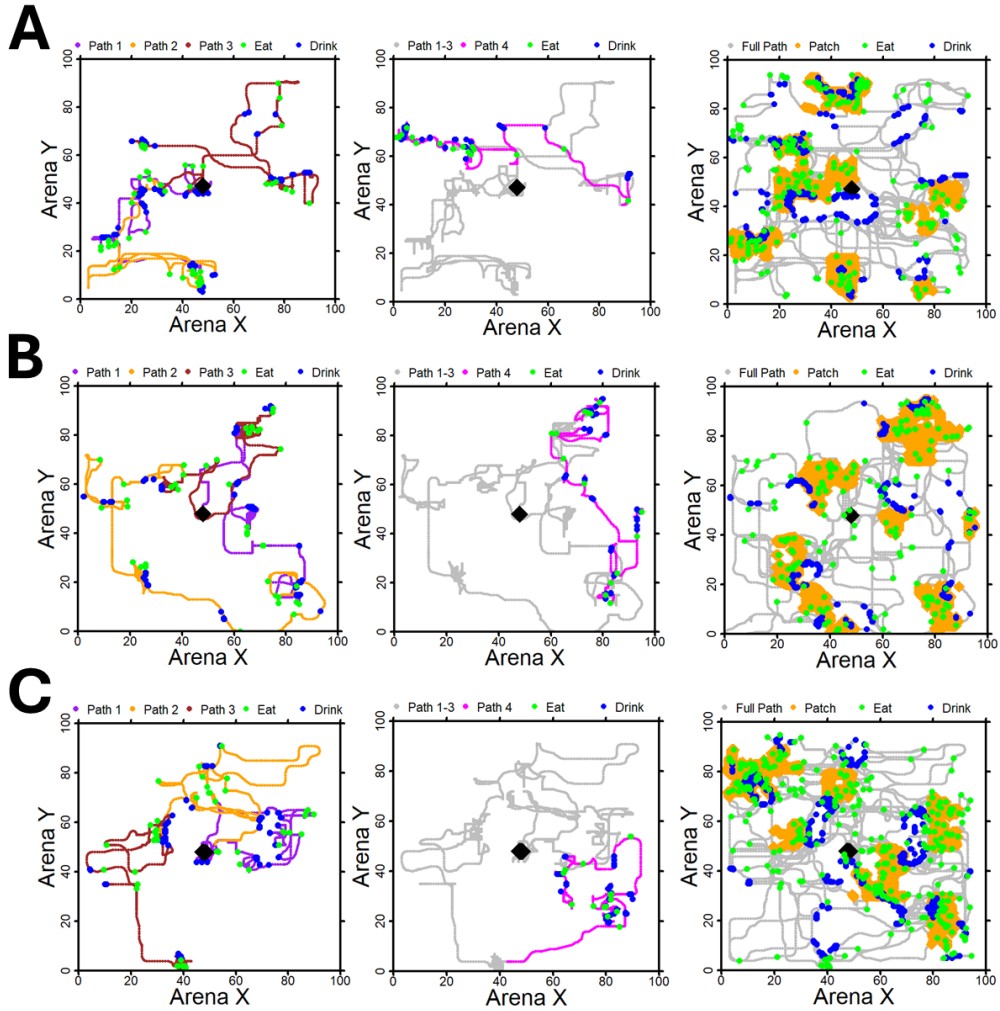

Figure 6: *Agents show stereotyped early exploration and sustained revisitation behavior in novel environments.* Representative trajectories from three episodes. Each row (A–C) illustrates three sequential stages from a single episode: early exploratory loops (left), a path sampled during the transition to revisitation (middle), and the full episode trajectory with revisited patch regions highlighted in orange (right). These examples demonstrate consistent exploratory patterns—such as outwardly expanding loops—and show that exploration persists even during revisitation, often aiding predator avoidance. This figure complements the summary view in Figure 1.

## B.5 Training Dynamics and Entropy Evolution

To better understand how behavior evolves over training, we tracked the average policy entropy, Figure 9 shows how decision confidence increases over time, as reflected in the entropy dynamics that indicate increasing policy stability and confidence in selecting goal-directed actions.

These dynamics complement the behavioral and representational changes reported in section 4, further reinforcing the conclusion that high-level behavioral structure and planning capacity emerge progressively during training.

## B.6 Memory and Predator-Related Ablations

To ensure our results were not driven by incidental dynamics or shallow heuristics, we conducted ablation experiments targeting memory mechanisms and predator response strategies. Figure 10

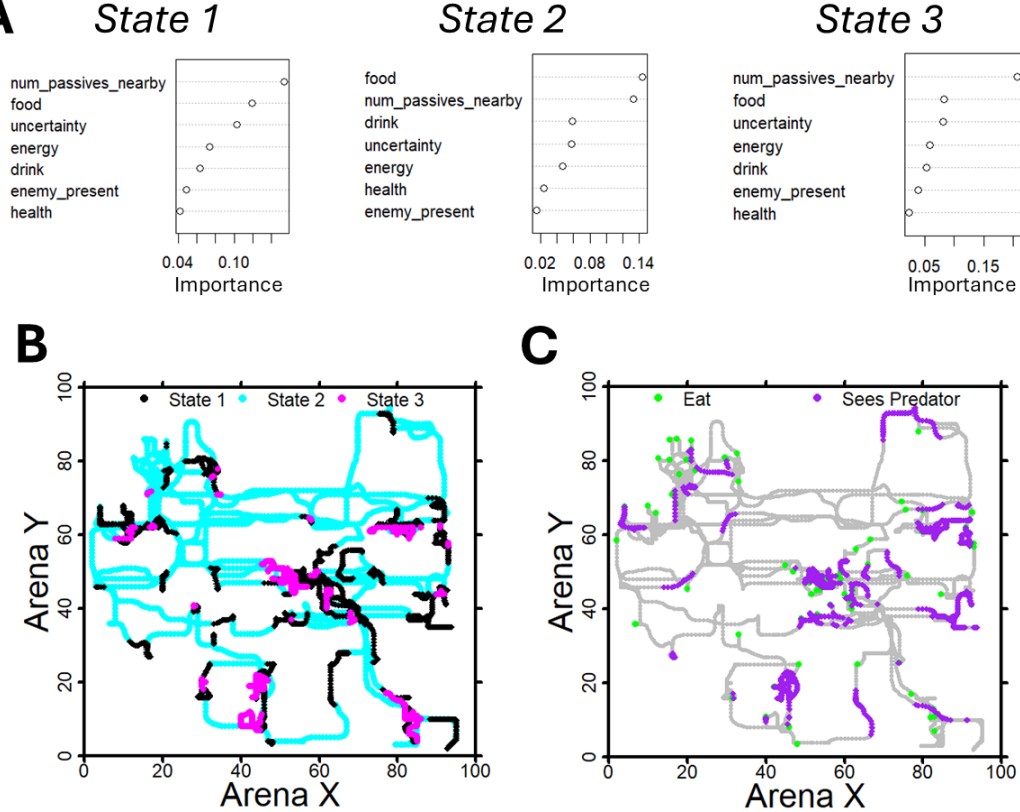

Figure 7: *Agents exhibit distinct movement patterns aligned with task variables such as food, fatigue, and predator presence.* (A) Variable importance from conditional inference trees [96] predicting transitions between three movement states identified via unsupervised clustering [95]: short-, mid-, and long-range locomotion. Key predictors include hunger level (*food*), positional uncertainty (*uncertainty*), predator presence (*enemy_present*), and nearby cow density (*num_passives_nearby*). (B) A representative episode segment colored by inferred movement state. (C) The same trajectory overlaid with predator encounters and eat events. State 1 (mid-range) frequently coincides with predator presence, while state 3 (short-range) clusters near food and eat actions. These results suggest that DRL agents flexibly shift between movement regimes in response to internal drives and environmental context—without requiring weight updates as these results were explored on test arenas with fixed weights.

shows a representative trajectory from a PPO agent without recurrence (i.e., no memory). These agents became trapped in inefficient local loops and had markedly reduced survival times, highlighting the critical role of memory in supporting long-horizon foraging.

In a second ablation, we ablated the auxiliary position-prediction objective (path integration). As shown in Figure 11, this had little effect on observed movement patterns. However, it significantly degraded decoding performance and eliminated structured spatial representations (see Figure 17 in the neural analysis section).

Lastly, we tested a predator-blind variant in which agents could not damage predators. As shown in Figure 4, agents exhibited similar movement strategies and achieved comparable survival times, indicating that predator evasion—not combat—is the primary learned strategy.

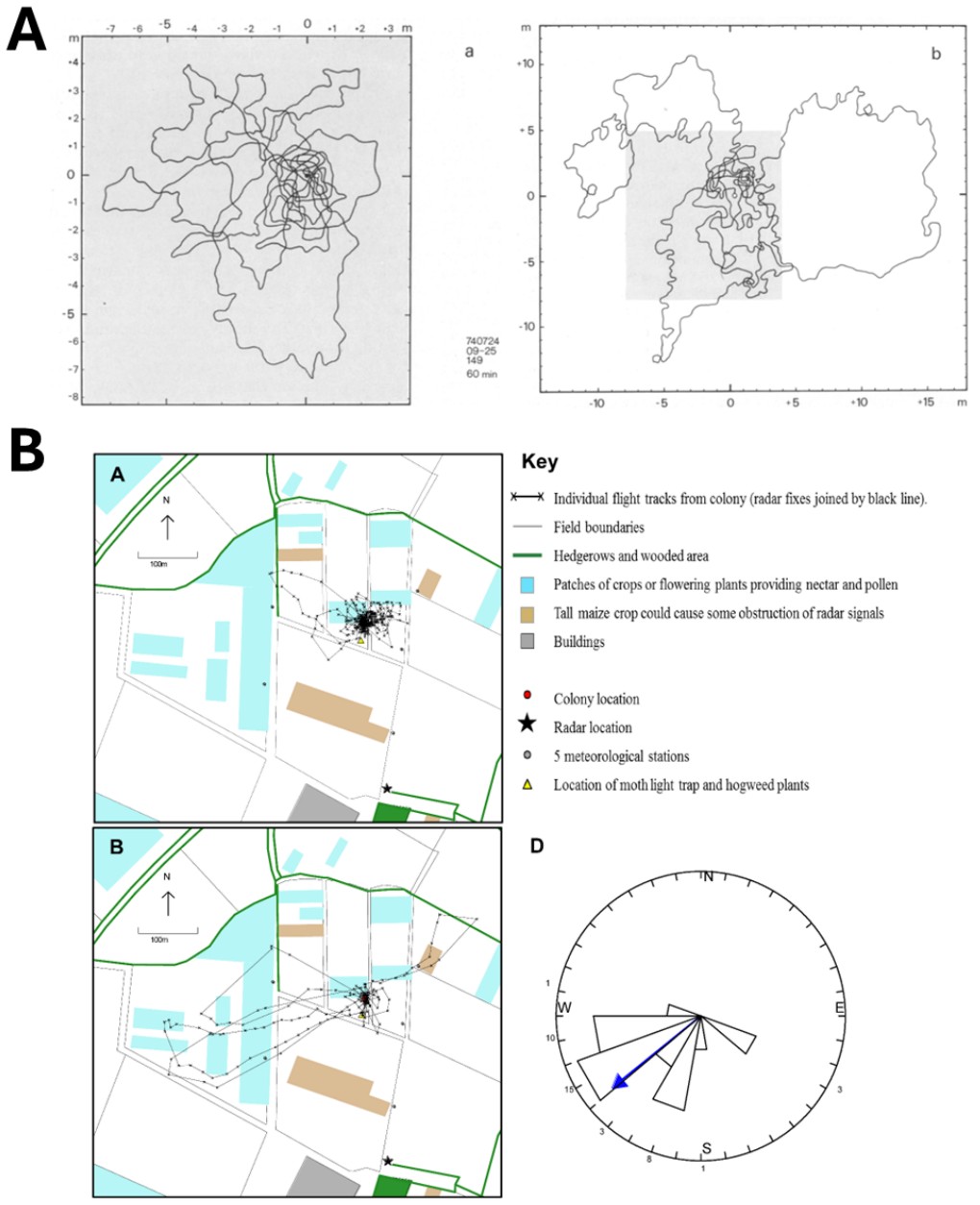

Figure 8: *Agent exploration trajectories qualitatively resemble real-world search patterns in ants and bees.* (A) Adapted from [79]: an ant's search trajectory following nest displacement, showing outwardly expanding loops over time. (B) Adapted from [97]: bee trajectories during their first, second, and third departures from the nest. In both cases, the animals exhibit looping paths that rotate azimuthally and expand in spatial scale—closely matching the trajectories observed in DRL agents during early foraging. These parallels suggest that similar search strategies may emerge across biological and artificial systems when operating under uncertainty and partial observability.

## C  Interpreting Agent Representations: Supplementary Analyses

This section presents supplementary analyses and figures that support the neural decoding results in section 4. We examine how pruning and auxiliary objectives influence spatial encoding, and provide additional insight into the structure and modularity of the learned internal representations.

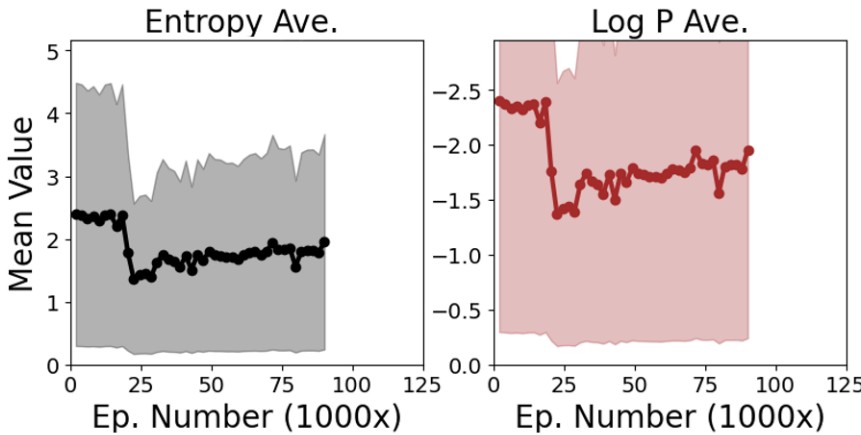

Figure 9: *Entropy dynamics reveal a transition from undirected exploration to confident, goal-directed behavior.* Evolution of average entropy over training for PPO-RNN agents. As training progresses, entropy declines and the log probability of the selected actions increases, reflecting increased confidence and policy refinement. Entropy slowly increases later in training as the agent discovers ways to maximize both task performance and policy entropy, reflective of the loss function.

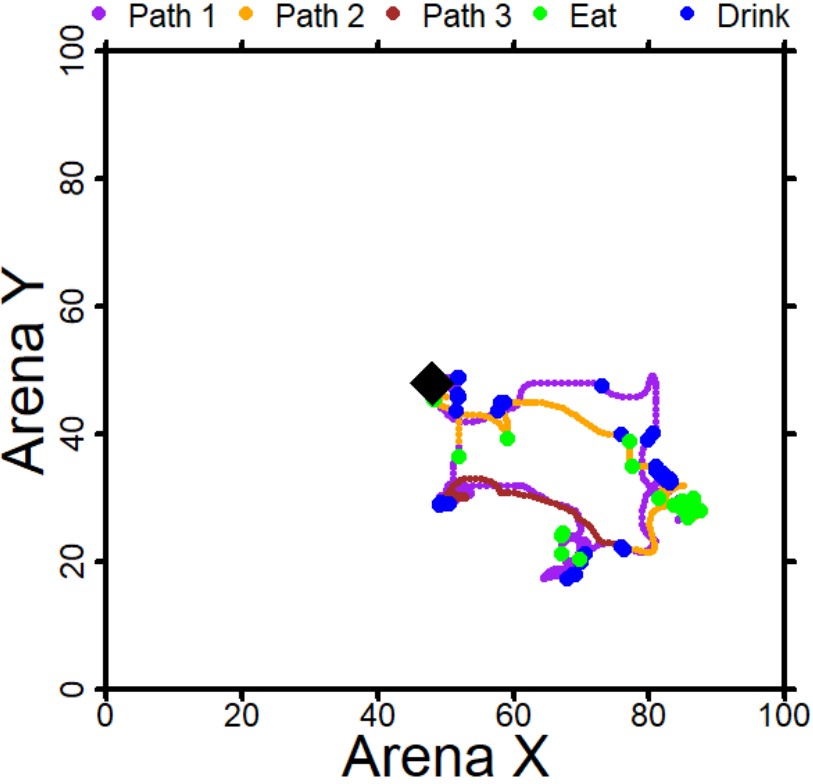

Figure 10: *Agents without recurrent memory exhibit poor spatial coverage and reduced survival.* Trajectory from a PPO agent without a recurrent architecture. Memoryless agents often became trapped in local loops and survived significantly fewer timesteps than agents with recurrence.

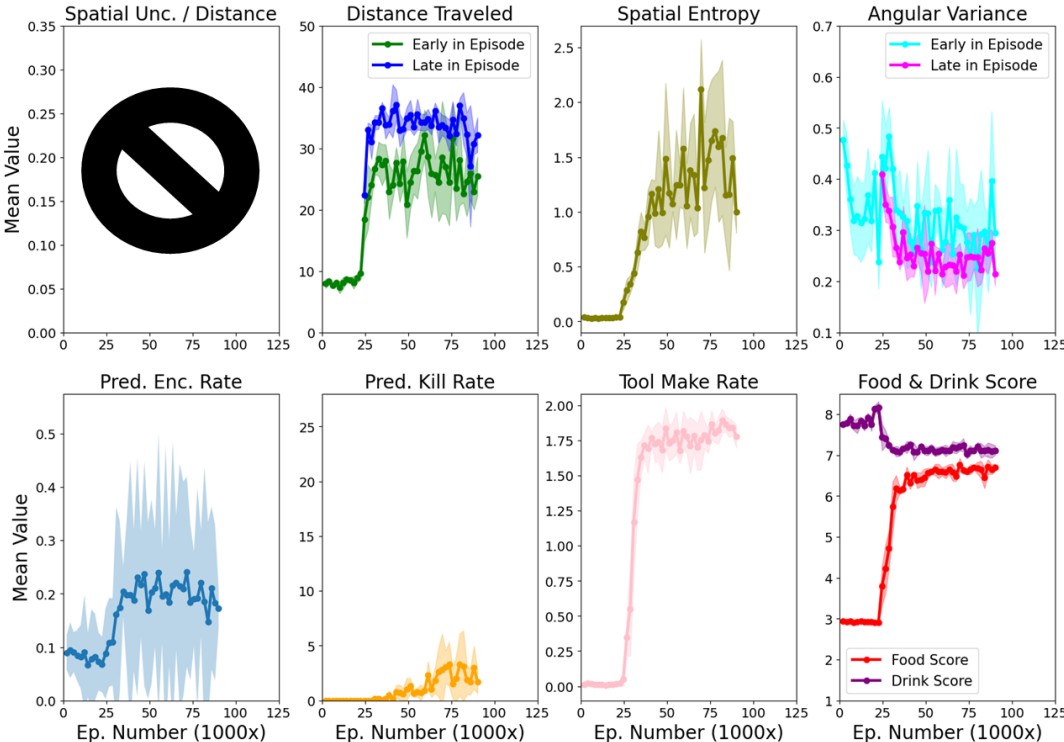

Figure 11: *Disabling the auxiliary position-prediction objective limits the predator killing abilities of agents.* Training dynamics and movement patterns for PPO-RNN agents without the auxiliary path integration objective. The absence of structured spatial uncertainty plots (top left) is due to the removal of the predicted position output normally produced by the auxiliary loss, since predicted current position is not outputted by the network in this experiment. The primary difference in learning history dynamics relative to the baseline model is that without the position-prediction objective, the emergence of predator killing abilities of agents is worse. This ability requires agents crafting a tool, with a sword being the highest damage tool, and reducing the predator health to 0. Predator killing is difficult because the predators can also kill the agent.

## C.1 Impact of Auxiliary Objectives on Decoding

Figure 17 shows that removing the auxiliary path integration objective eliminated above-chance decoding of allocentric position from the RNN state. This indicates that the auxiliary loss was not merely a regularizer, but essential for inducing spatially interpretable internal representations.

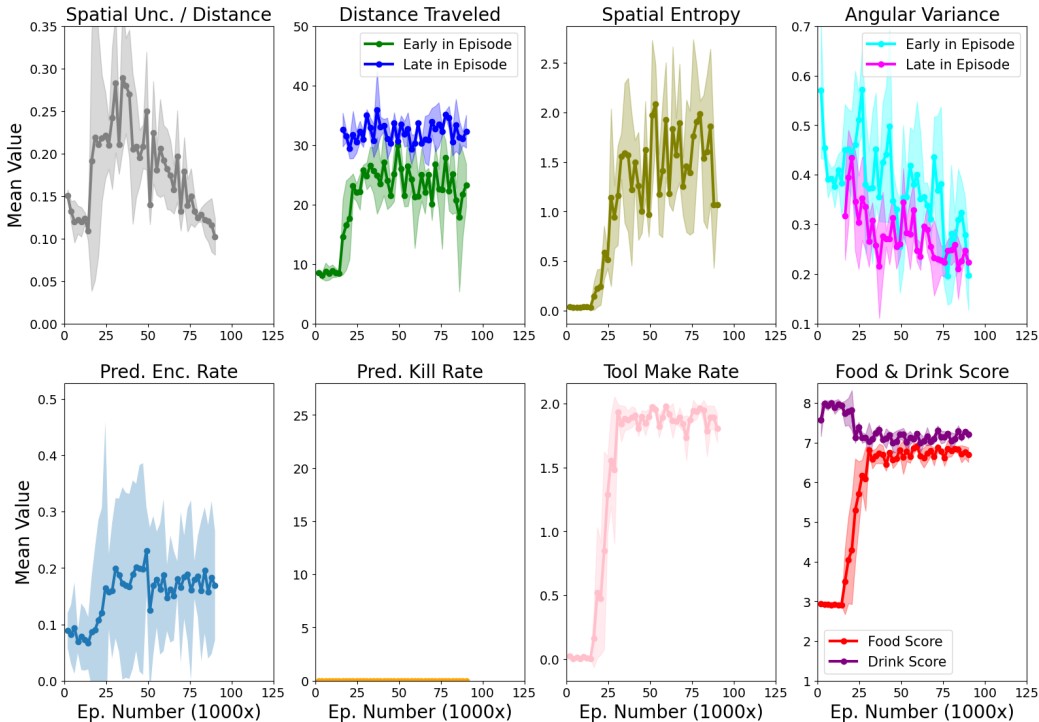

Figure 12: *Predator evasion dominates over combat in learned foraging behavior.* PPO-RNN agents trained without the ability to damage predators exhibited similar movement patterns and survival outcomes. This indicates that effective predator evasion—rather than combat—is the primary strategy developed by the agent.

**A**

```
Generalized Linear Model Regression Results
==============================================================================
Dep. Variable:            PatchRevisit   No. Observations:                 7978
Model:                             GLM   Df Residuals:                     7966
Model Family:                 Binomial   Df Model:                           11
Link Function:                   Logit   Scale:                          1.0000
Method:                           IRLS   Log-Likelihood:                -5019.4
Date:                 Wed, 14 May 2025   Deviance:                        10039.
Time:                         18:28:57   Pearson chi2:                 9.85e+03
No. Iterations:                      4   Pseudo R-squ. (CS):             0.1201
Covariance Type:             nonrobust
==============================================================================
                   coef    std err          z      P>|z|      [0.025      0.975]
------------------------------------------------------------------------------
Intercept       -0.0063      0.055     -0.114      0.909      -0.114       0.102
C(AgentID)[T.1]  0.0512      0.070      0.728      0.466      -0.087       0.189
C(AgentID)[T.2] -0.0758      0.074     -1.019      0.308      -0.222       0.070
C(AgentID)[T.3]  0.0125      0.090      0.138      0.890      -0.165       0.190
C(AgentID)[T.4]  0.0780      0.081      0.963      0.336      -0.081       0.237
EatRate         -0.0804      0.030     -2.652      0.008      -0.140      -0.021
DrinkRate        0.0071      0.030      0.241      0.809      -0.051       0.065
PredRate        -0.0928      0.027     -3.434      0.001      -0.146      -0.040
Recency         -0.5980      0.029    -20.811      0.000      -0.654      -0.542
Dwelltime       -0.0635      0.030     -2.151      0.032      -0.121      -0.006
CowCount         0.5308      0.032     16.504      0.000       0.468       0.594
Uncertainty      0.0946      0.025      3.761      0.000       0.045       0.144
==============================================================================
```

**B**

```
         feature       VIF
0        EatRate  1.464861
1      DrinkRate  1.472354
2       PredRate  1.164214
3        Recency  1.031999
4      Dwelltime  1.462161
5       CowCount  1.259648
6    Uncertainty  1.013469
Statistic 0.8856782829636347
p-value 1.9469406301235196e-60
LM Statistic 4396.655906048893
LM-Test p-value 0.0
F-Statistic 144.9366132425643
F-Test p-value 0.0
```

Figure 13: *GLM results confirm that patch revisitation decisions are shaped by uncertainty, reward history, and predator presence.* (A) Full GLM output underlying Figure 3, obtained using `statsmodels.formula.api` in Python. The model classified patches as chosen (1) or not (0) based on average historical variables, using agent ID as a fixed effect. A single model was fit jointly across five PPO-RNN agents, spanning 7978 revisitation decisions. (B) Variance inflation factor (VIF) scores show no excessive multicollinearity among predictors (all VIF < 10).

Table 2: *Variables logged at each timestep to support downstream behavioral and neural analyses.* These include internal state variables, agent decisions, environmental context, predictive outputs, and metadata, and are used to compute all behavioral metrics and decoding analyses reported in the paper.

| Variable | Description |
|---|---|
| Action | Action selected by the agent |
| Health | Current health of the agent |
| Food | Current food level of the agent |
| Drink | Current water level of the agent |
| Energy | Current energy level of the agent |
| Done | Did the episode end? |
| Is Sleeping | Is the agent currently sleeping? |
| Is Resting | Whether the agent is in a resting state (disabled in all experiments presented) |
| Player Position | Agent's absolute X/Y position relative to the top-left corner of the 96×96 arena (origin at 0,0) |
| Recover | The current state of the timer for the agent to recover a point of health |
| Hunger | The current state of the timer for the agent to lose a point of food |
| Thirst | The current state of the timer for the agent to lose a point of drink |
| Fatigue | The current state of the timer for the agent to lose a point of energy |
| Light Level | The light level variable, which modifies predator spawn rates (higher when lower, i.e. at night) |
| Distance to Melee | L1 distance to the nearest melee predator |
| Melee on Screen | Is there a melee predator currently on screen? |
| Distance to Passive | L1 distance to the nearest cow/food animal |
| Passive on Screen | Is there a cow currently on screen? |
| Distance to Ranged | L1 distance to the nearest ranged predator |
| Ranged on Screen | Is there a ranged predator currently on screen? |
| Num Melee Nearby | How many melee predators are currently on screen? |
| Num Passives Nearby | How many cows are currently on screen? |
| Num Ranged Nearby | How many ranged predators are currently on screen? |
| Delta X & Y (Relative Position) | Agent's position relative to its starting location (center of arena) |
| Predicted Delta X & Y | Agent's internal prediction of its position relative to its starting location |
| Num Monsters Killed | How many predators has the agent defeated this episode so far? |
| Has Sword | Whether the agent has crafted a sword to improve combat effectiveness this episode |
| Has Pick | Whether the agent has crafted a pickaxe to remove stone obstacle tiles |
| Held Iron | Number of iron resources collected (used to craft improved weapons) |
| Value | The agent's value function output |
| Entropy | The entropy of the agent's policy for the current timestep |
| Log Probability | Log probability of the selected action under the policy at the current timestep |
| Episode ID | Random ID for the current episode |

Table 3: Model and environment hyperparameters used for the *baseline* configuration.

| Parameter | Default Value | How Chosen (if applicable) |
|---|---|---|
| **Model Parameters** | | |
| Learning Rate | 0.0002 | Craftax default |
| $\gamma$ | 0.99 | " |
| $\lambda_{\text{GAE}}$ | 0.8 | Same as Craftax default |
| PPO clipping $\epsilon$ | 0.2 | Same as Craftax default |
| $\mathcal{W}_{\text{V}}$ | 0.5 | Same as Craftax default |
| $\mathcal{W}_{\text{entropy}}$ | 0.01 | Same as Craftax default |
| Activation function | tanh | Same as Craftax default |
| Number of neurons per layer | 512 | Same as Craftax default |
| Steps per PPO iteration | 64 | Same as Craftax default |
| Number of parallel environments | 1024 | Same as Craftax default |
| Number of epochs per PPO iteration | 4 | Same as Craftax default |
| Number of minibatches per epoch | 8 | Same as Craftax default |
| Total training timesteps | 3,000,000,000 | Chosen to ensure convergence or trend visibility within a 24–48 hour wall-clock training window |
| $\mathcal{W}_{\text{aux}}$ | 0.025 | Parameter sweep of {0.01, 0.1, 0.025, 1.0}, balancing performance and position encoding quality |
| Pruning type | Magnitude | Only JaxPruner type that did not severely degrade performance in testing |
| Prune Step | 20,000 | Mid-training: after baseline agent begins performing reasonably, but before convergence |
| **PQN-specific Parameters** | | |
| $\epsilon$ Start | 1.0 | PQN default |
| $\epsilon$ Finish | 0.05 | Parameter sweep of {0.005, 0.01, 0.05, 0.1} |
| $\epsilon$ Decay | 1.0 | Parameter sweep of {0.1, 1.0, 2.0} |
| Total training timesteps | 12,000,000,000 | Chosen to allow PQN time to explore+converge |
| Total decay timesteps | 6,000,000,000 | Parameter sweep of {1B, 6B, 12B} |
| Num environment steps per update | 128 | PQN default |
| $\lambda$ | 0.5 | PQN default |
| **Environment Parameters** | | |
| Predators | True | |
| Max Cows | 108 | Selected from a sweep {48, 72, 108} to balance patch-leaving and revisitation in competent agents |
| Use full action space | False | Disabled actions not relevant to ForageWorld (subset of original Craftax action space) |
| Map Size | 96 | Chosen from among {24, 48, 96} to maximize the navigation and memory challenge for the agent |
| Directional Vision | False | |

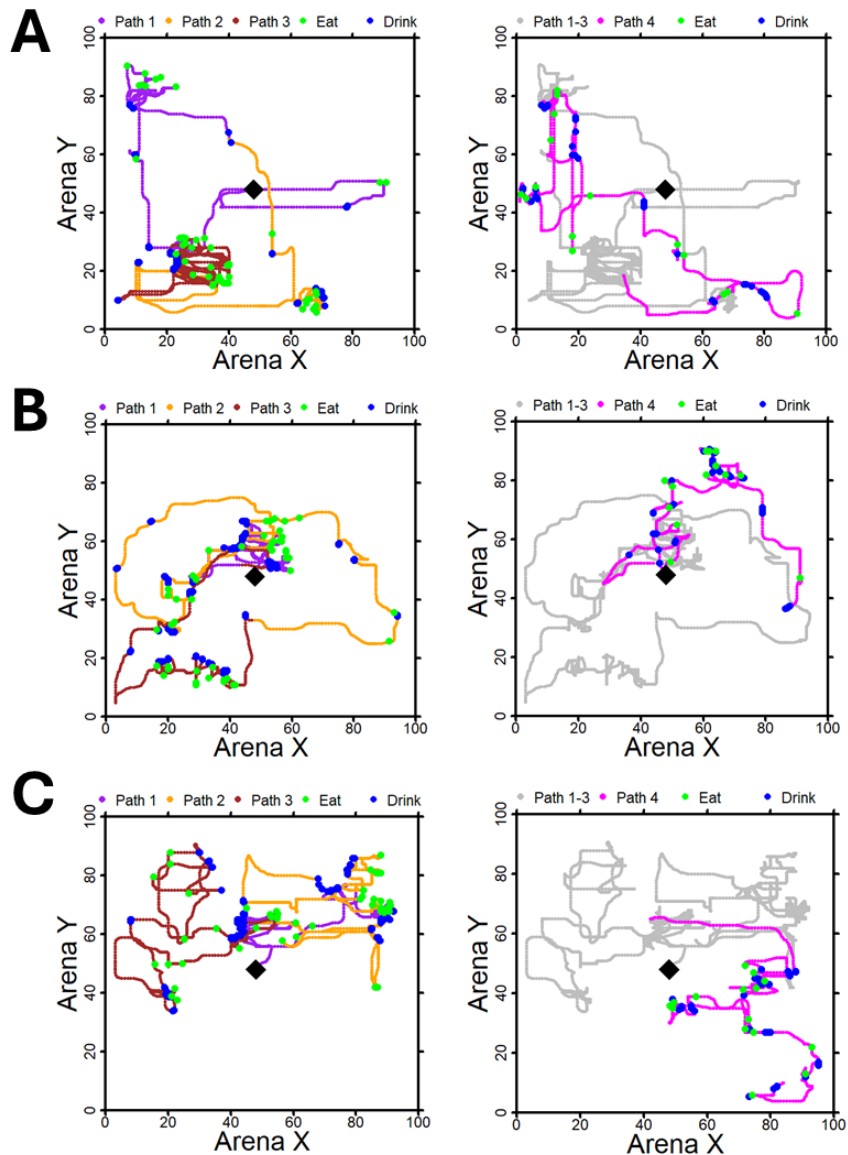

Figure 14: *Front-facing agents exhibit denser local exploration in early training.* Each row shows a representative early-exploration trajectory for an agent with a front-facing field of view (FOV), as opposed to the 360° egocentric view used in the baseline configuration. Front-facing agents tended to perform intermittent localized and dense exploratory behavior during the early phases of exploration.

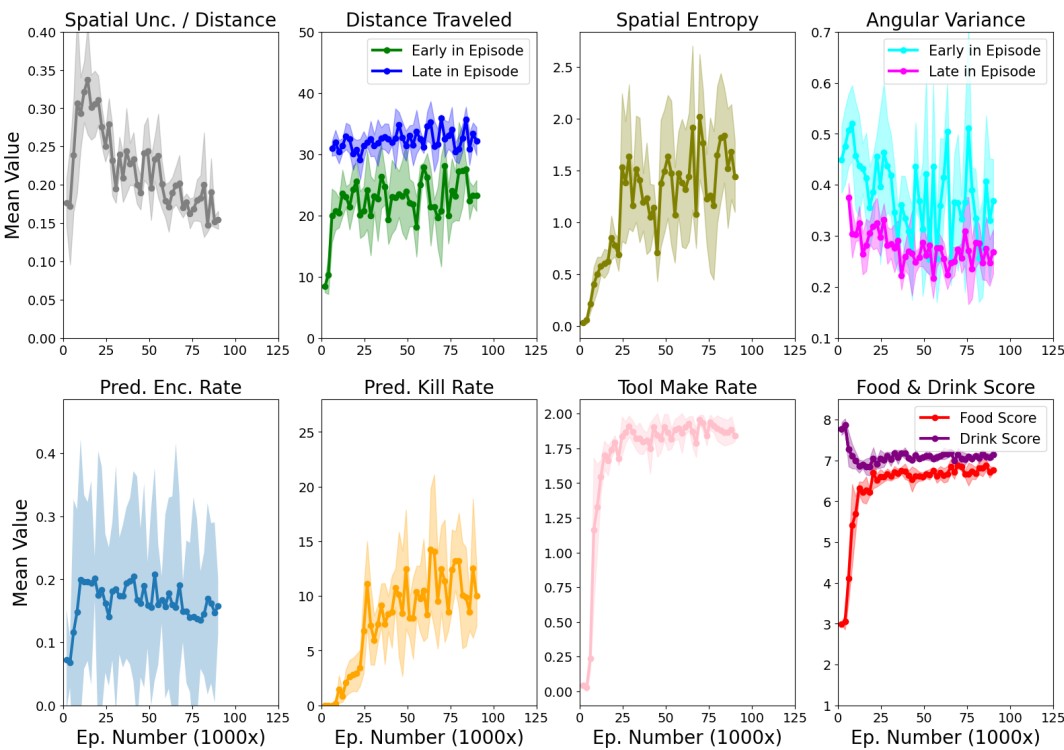

Figure 15: *Front-facing FOV agents exhibit earlier onset of long-range exploration.* Variant of Figure 4 for PPO-RNN agents with a front-facing field of view. These agents showed a faster shift toward long-range exploration compared to baseline agents. This modification may serve as a useful manipulation in future studies of sensory constraints.

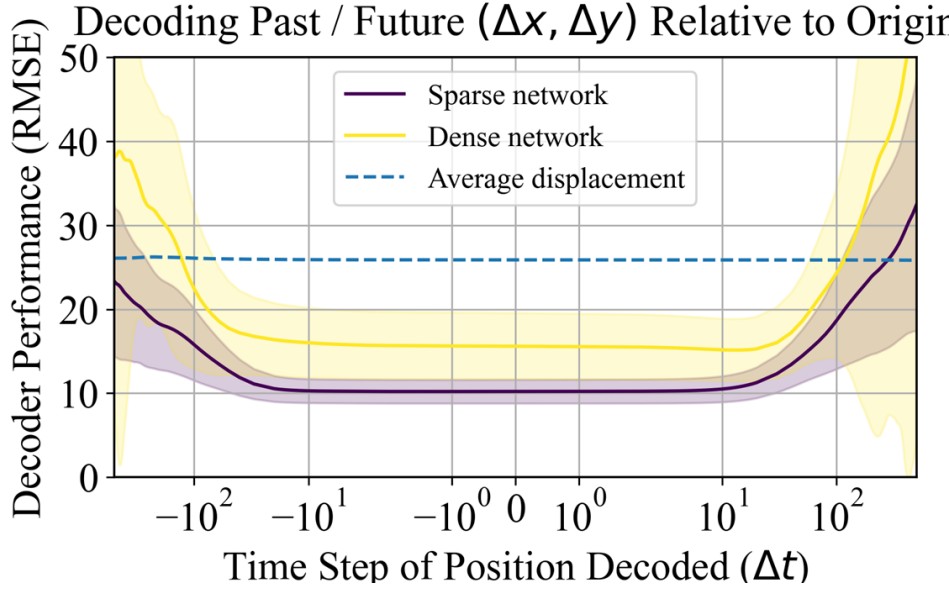

Figure 16: *Pruned networks yield better allocentric position decoding than fully connected networks.* Comparison of allocentric position decoding accuracy (past/future) between sparse (90%) and unpruned PPO-RNN agents. Sparse models show higher accuracy, potentially due to more compact and modular encoding of task-relevant variables. Error bars show 95% confidence intervals.

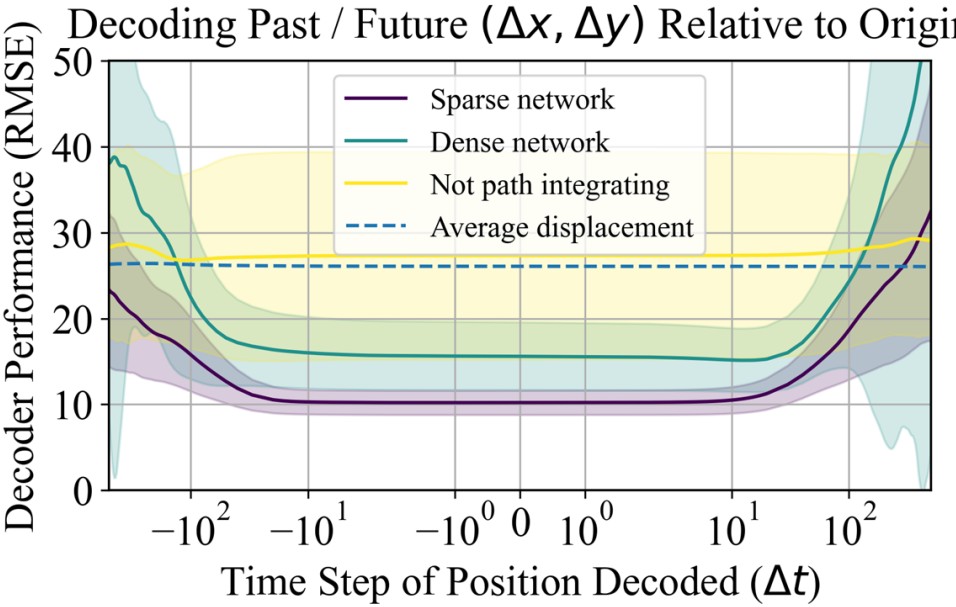

Figure 17: *The auxiliary position-prediction objective is necessary for spatially interpretable representations.* Without the auxiliary path integration loss, allocentric position decoding from the agent's RNN state drops to chance levels—eliminating the structured spatial encoding observed in the baseline model. Bars show decoding accuracy for past and future positions; error bars reflect 95% confidence intervals.

## C.2 Coefficient Structure and Functional Modularity

To better understand how position information is distributed across neurons, we analyzed the regression coefficients learned by decoders trained to predict past and future positions. Figure 18 visualizes these ridge regression weights, aligned by neuron ID, for both sparse and dense PPO-RNN agents. In both architectures, overlapping neuron populations contributed to past and future decoding, but this overlap diminished at longer temporal offsets. Sparse networks exhibited clearer separation between past- and future-coding units, as well as higher overall sparsity in their weight maps—factors that may underlie their superior decoding performance.

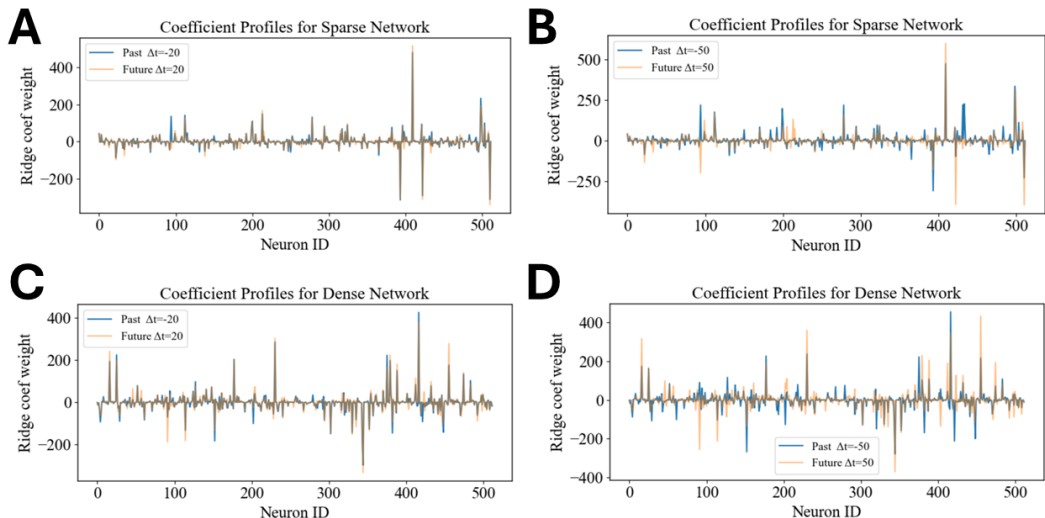

Figure 18: *Sparse agents exhibit modular and separable encoding of past and future spatial positions.* Ridge regression weights for allocentric position decoding, aligned by neuron ID. (A, B) show sparse networks at 20- and 50-timestep prediction horizons; (C, D) show dense networks. While both architectures use overlapping units for past and future decoding, sparse networks show greater separation at longer horizons—consistent with their improved decoding performance.

## C.3 Regularization and Train-Test Split for Decoding

Some RNN units remain inactive for entire episodes, making the decoding problem potentially ill-conditioned. Accordingly, we applied ridge regression with an $\ell_2$ regularization term:

$$L_{\Delta t}(f) = \sum_{i=1}^{N} \left( Y_{t+\Delta t}^i - f(h_t^i) \right)^2 + \alpha \|f\|_K^2$$

This regularization improves model stability while preserving interpretability. A separate decoder was trained for each prediction offset $\Delta t$ using the following linear model:

$$f(h_t) = Ah_t + b, \quad A \in \mathbb{R}^{2 \times 512}, \quad b \in \mathbb{R}^2$$

We selected ridge regression for its favorable tradeoffs among interpretability, numerical stability, and computational efficiency. With time complexity $O(Np^2)$ and direct access to regression coefficients, this approach enables fine-grained analysis of how individual RNN units contribute to spatial encoding.

### Train-Test Split Design

To prevent information leakage due to temporal continuity in $h_t$, we employed a structured train-test split. Each decoder was trained on the first 75% of timesteps within an episode and evaluated on the remaining 25%. This design ensures that models cannot rely on short-range temporal correlations and must generalize across broader regions of feature space. All decoders were trained across multiple episodes to ensure robustness to variation in arena layouts and agent trajectories.

## D   Generalized linear model for predicting neural activity from current position

As a supplementary analysis, we explored how position was represented at the single neuron level in the RNNs. NeMoS (Neural ModelS) was used for fitting generalized linear models (GLMs) that predicted neural activity from the current position of the agent, see https://github.com/flatironinstitute/nemos for extended documentation [98]. One GLM was fit per neuron, but multiple episodes (arena configurations) were used for each model fit, typically 5-10 episodes minimum to avoid overtraining to one arena configuration. Prior to setting up the model, the 96 x 96 arena was coarse-grained into 14 x 14 position bins, and a binary variable per bin was encoded that reported whether or not the agent was in a given bin (196 total regressor coefficients). Neural data was also shifted into the positive axis (lowest value zeroed), and normalized to the range of 0-1 per neuron to extract the activity changes relative to a given neuron's baseline for easier cross-neuron comparison.

The GLM fit the first 70% of each episode in the training set of episodes Figure 19. The majority of neurons fit had comparable performance between the test and training sets Figure 20.

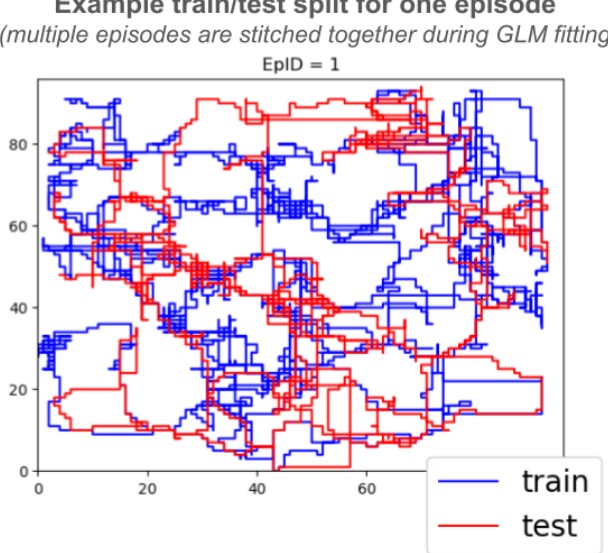

Figure 19: *Example test/train split for a single episode to be used in GLM fitting.* GLM models, one model per neuron, with 5-10 episodes per model, predicted neural activity from position. The first 70% of an episode was trained, while the last 30% was the test split.

Approximately 100/512 neurons had good fit performance using just position regressors across runs, with values typically in the range of 60-120 position-sensitive neurons per training run. An example good fit is shown in Figure 21.

We then plotted the number of neurons with GLM coefficients in the three largest mean-valued k-means clusters out of five clusters, see Figure 22 for the selection. We also plotted the average GLM coefficient per position bin Figure 23. The number of strongly position encoding neurons increased with radial distance from the origin of the arena, and the average GLM coefficient magnitude also increased with radial distance from the origin Figure 24. This trend was seen in all main conditions, PPO-RNN with and without path integration, and PQN-RNN. Thus, an accumulation circuit may ramp with distance from origin and pressure the agent to return to the origin the farther away it gets, partially explaining the periodic origin revisiting behavior observed across conditions. This pattern was present even without the position predicting auxiliary objective Figure 25 and Figure 26, and also in the PQN-RNN runs Figure 27 and Figure 28, suggesting it is fundamental to the task solution. Example scripts to use this analysis method are in the github repository for this paper.

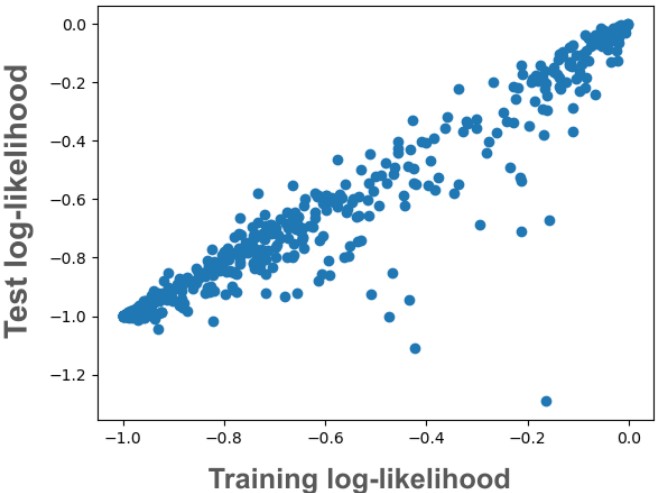

Figure 20: *Log likelihood scores* for the same GLM model (single neuron fit), showing the difference in log likelihood between test and train data, one model per data point. The data here is a PPO-RNN baseline run.

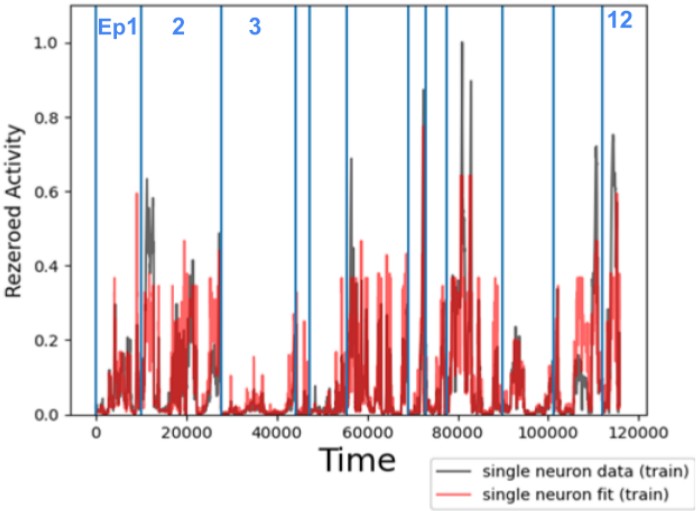

Figure 21: *Example of a single neuron, multi-episode model fit.* Each of the 512 GLM models fit, one per neuron, has one GLM coefficient for each of 196 position bins (14 x 14) that the arena grids were coarse-grain binned into for this analysis. We show an example good fit for a single neuron's data (black), fit to 12 different episodes for one model (fit in red).

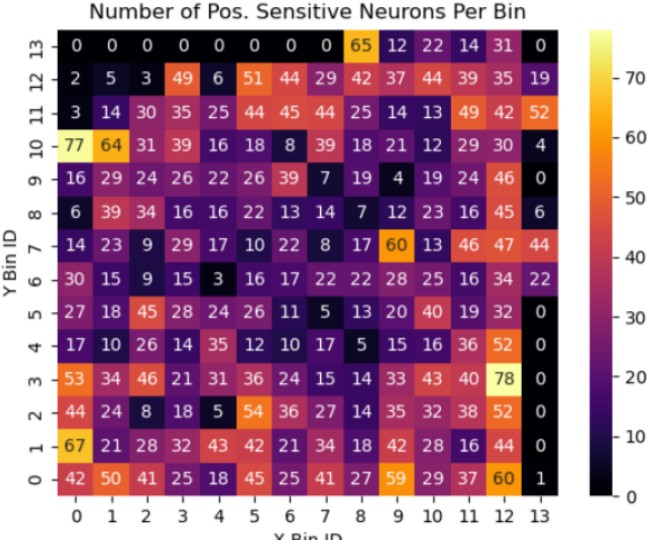

Figure 22: *Number of position-sensitive neurons per position bin in the arena for PPO-RNN.* Each of the 512 GLM models fit, one per neuron, has one GLM coefficient for each of 196 position bins (14 x 14) that the arena grids were coarse-grain binned into for this analysis. The heatmap of the number of well-fit neurons per position bin is shown in the plot. The data here is a PPO-RNN baseline run. The 0's in the uppermost row and rightmost column of bins are due to the agents never going out to those position bins in these runs.

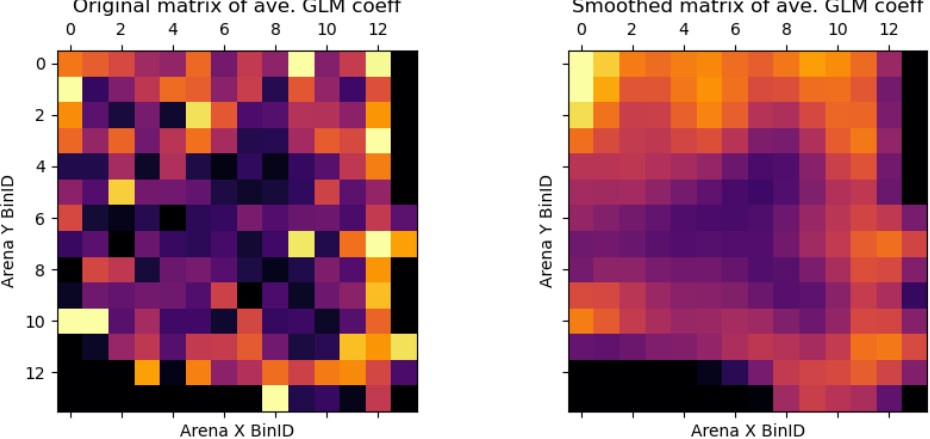

Figure 23: *Heatmap of average GLM coefficient magnitude per position bin for PPO-RNN.* Each of the 512 GLM models fit, one per neuron, has one GLM coefficient for each of 196 position bins (14 x 14) that the arena grids were coarse-grain binned into for this analysis. These plots show the average GLM coefficient value across well-fit neurons per bin (left), and the Gaussian smoothed shape of the same heatmap (right). The data here is a PPO-RNN baseline run. The 0's in the lowermost row and rightmost column of bins are due to the agents never going out to those position bins in these runs.

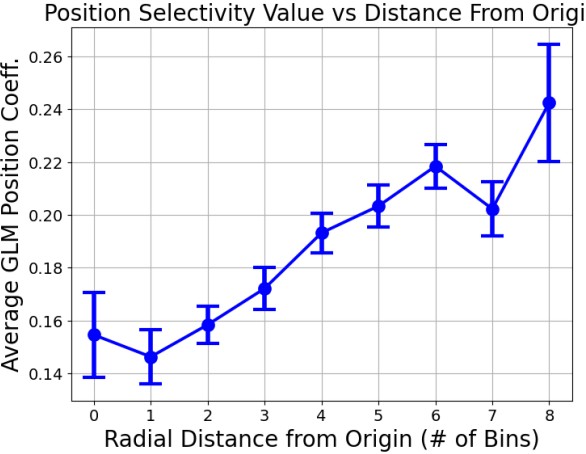

Figure 24: *Average position bin GLM coefficient versus radial distance from the arena origin (center) for PPO-RNN.* Each of the 512 GLM models fit, one per neuron, has one GLM coefficient for each of 196 position bins (14 x 14) that the arena grids were coarse-grain binned into for this analysis. The average GLM coefficient for the well-fit neurons per radial bin distance from the origin was plotted, finding the neural response to a position change increases with distance from the origin. The data here is a PPO-RNN baseline run. Data points are the mean across neurons while error bars are 95% CI.

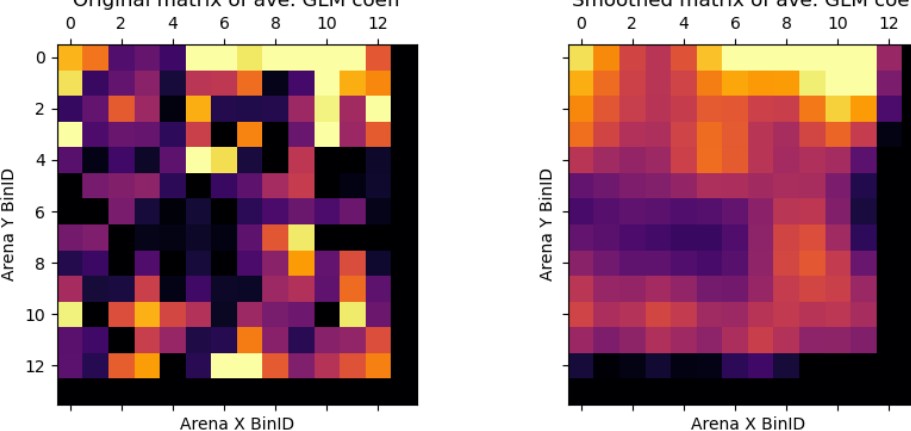

Figure 25: *Heatmap of average GLM coefficient magnitude per position bin for PPO-RNN without path integration.* Each of the 512 GLM models fit, one per neuron, has one GLM coefficient for each of 196 position bins (14 x 14) that the arena grids were coarse-grain binned into for this analysis. These plots show the average GLM coefficient value across well-fit neurons per bin (left), and the Gaussian smoothed shape of the same heatmap (right). The data here is a PPO-RNN without position-prediction (no path integration) run. The 0's in the lowermost row and rightmost column of bins are due to the agents never going out to those position bins in these runs.

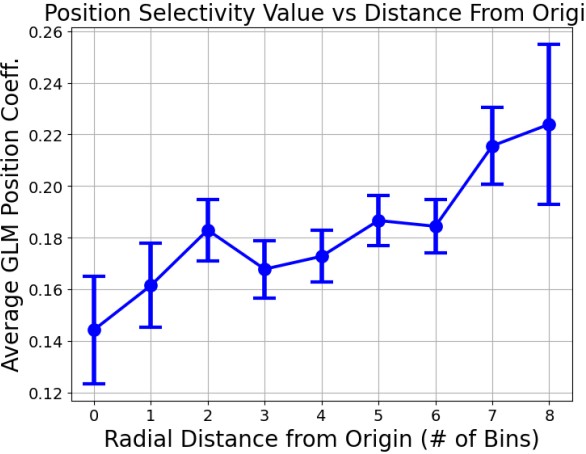

Figure 26: *Average position bin GLM coefficient versus radial distance from the arena origin (center) for PPO-RNN without path integration.* Each of the 512 GLM models fit, one per neuron, has one GLM coefficient for each of 196 position bins (14 x 14) that the arena grids were coarse-grain binned into for this analysis. The average GLM coefficient for the well-fit neurons per radial bin distance from the origin was plotted, finding the neural response to a position change increases with distance from the origin. The data here is a PPO-RNN without position-prediction (no path integration) run. Data points are the mean across neurons while error bars are 95% CI.

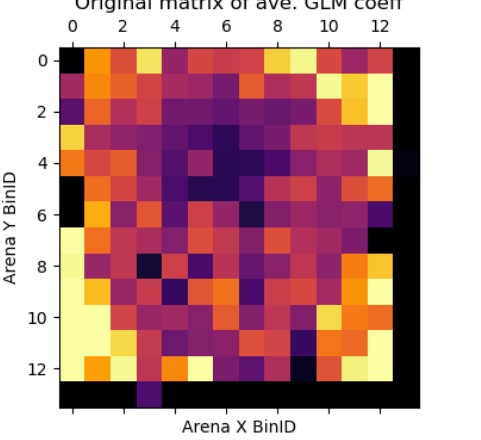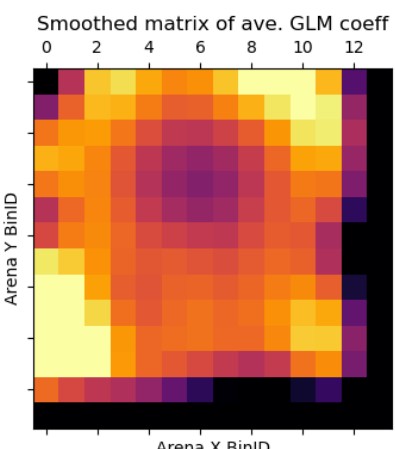

Figure 27: *Heatmap of average GLM coefficient magnitude per position bin for PQN-RNN.* Each of the 512 GLM models fit, one per neuron, has one GLM coefficient for each of 196 position bins (14 x 14) that the arena grids were coarse-grain binned into for this analysis. These plots show the average GLM coefficient value across well-fit neurons per bin (left), and the Gaussian smoothed shape of the same heatmap (right). The data here is a PQN-RNN run. The 0's in the lowermost row and rightmost column of bins are due to the agents never going out to those position bins in these runs.

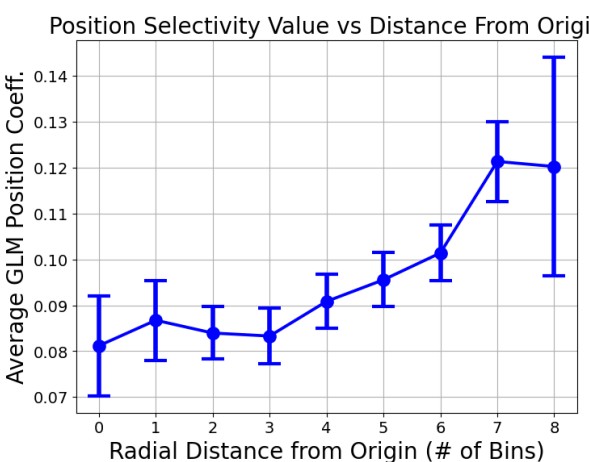

Figure 28: *Average position bin GLM coefficient versus radial distance from the arena origin (center) for PQN-RNN.* Each of the 512 GLM models fit, one per neuron, has one GLM coefficient for each of 196 position bins (14 x 14) that the arena grids were coarse-grain binned into for this analysis. The average GLM coefficient for the well-fit neurons per radial bin distance from the origin was plotted, finding the neural response to a position change increases with distance from the origin. The data here is a PQN-RNN run. Data points are the mean across neurons while error bars are 95% CI.

# E   Alternative DRL Objective: Parallelized Q-networks (PQN)

Recent work has shown that the off-policy RL algorithm parallelized Q-networks (PQN) can perform well in vanilla Craftax [75]. We explored whether this alternative DRL algorithm performs differently compared to PPO on ForageWorld. We adapted the PQN algorithm to ForageWorld using the LSTM model architecture and default hyperparameters from [75], which differ in some ways from our PPO implementation (for example, using an LSTM instead of a GRU for the RNN). Except where noted in Table 3, we used the same environment and training hyperparameters between PPO and PQN. This experiment provides an orthogonal test of ForageWorld– would a different algorithm, model, and training parameters result in different behavior?

We found that the PQN-LSTM models performed comparably to the PPO-GRU baseline for the subset of PQN models that learned the task (Figure 29), but that only a subset of PQN runs would converge to this level of performance– across the two successful test conditions that varied the exploration hyperparameters of PQN (in particular controlling the rate of decay of $\epsilon$, the chance of taking a random action instead of the optimal action under the current Q value function of PQN), only 30% of PQN runs performed comparably to the PPO model, with the remainder failing catastrophically in a way that was not seen in the baseline PPO runs. We believe this is the result of suboptimal exploration by PQN, i.e. the poorly performing PQN runs learned simple and reliable but suboptimal behaviors early in training and then were slow to deviate from them later on. For the runs that did perform comparably to PPO, we found overall similar learning histories and pathing behavior to the PPO baseline runs (Figure 30, Figure 31), but with the noticeable behavioral differences of much lower rates of predator killing (i.e. reducing predator HP to 0) (Figure 31) in the learning history analysis. Thus, this result suggests that, despite showing comparable overall performance at the task, PPO seems to converge to a predator fighting strategy, while PQN converges to a predator evasion strategy.

Figure 29: *Performance comparison for PQN versus PPO.* Individual PQN-LSTM runs performed comparably to the PPO-GRU baseline for two different exploration hyperparameter configurations, but the majority of runs failed to learn more than a trivial policy. Condition 1 used the $\epsilon$ decay parameters shown in Table 3, while condition 2 adjusted $\epsilon$ start, finish, and decay parameters so that $\epsilon$ decayed half as fast and ended at a final value of 0.1 (twice the chance of a random action). No condition resulted in uniformly successful runs. Other exploration hyperparameter values were tested, which are detailed in Table 3, but did not result in any high-performing runs.

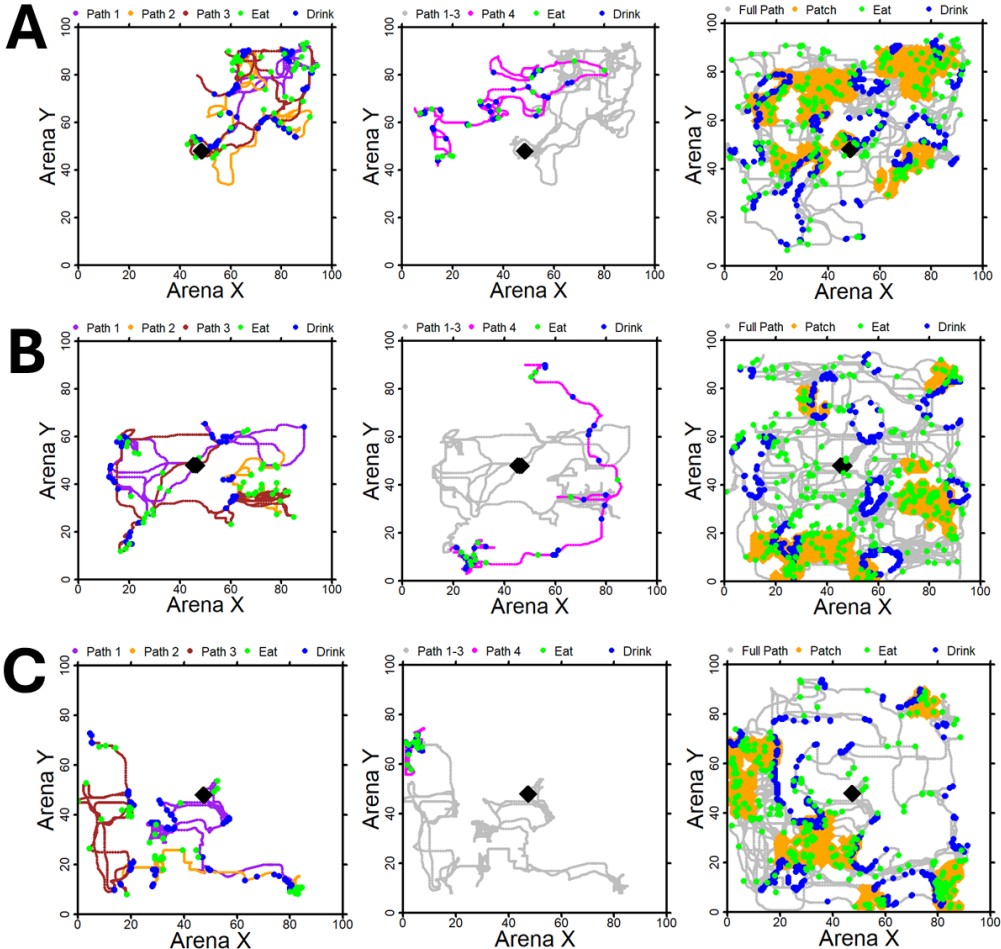

Figure 30: *Representative trajectories from three episodes for PQN-RNN.* Each row (A–C) illustrates three sequential stages from a single episode: early exploratory loops (left), a path sampled during the transition to revisitation (middle), and the full episode trajectory with revisited patch regions highlighted in orange (right).

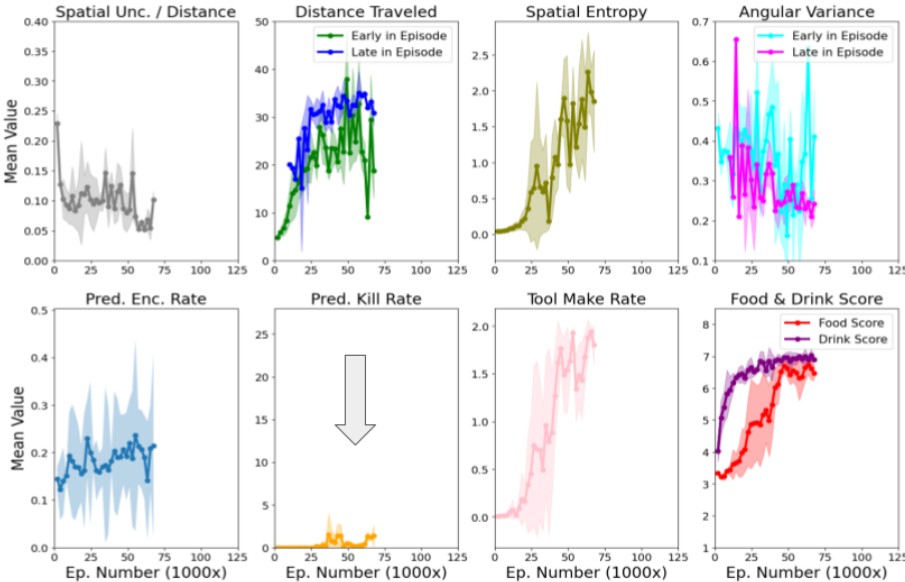

Figure 31: *Training dynamics and movement patterns learning history for PQN-RNN agents with the auxiliary path integration objective.* PQN had overall similar patterns to PPO, but was found to not learn predator killing as well (gray arrow).

# F  Toward Architectures for Cognitive Map Formation (Future Directions)

While our agents do not yet use structured maps, our findings suggest that such architectures may enhance both behavior and interpretability. Building on these findings about memory, planning, and spatial encoding in model-free agents, we highlight one architectural direction and outline broader opportunities for future work. An important next step is to explore architectures that more closely reflect mammalian spatial navigation capabilities. In biological systems, both allocentric and egocentric representations support planning and memory [99], particularly through grid-based spatial computations [100].

These "cognitive maps" [5, 101–104] extend beyond current position encoding—they support retrospection, prospective planning, and flexible decision-making [105–107]. Recent models have begun incorporating cognitive map-like representations into reinforcement learning frameworks [108].

In animals, spatial maps are modulated by task variables such as goals, landmarks, and reward-related events [109, 110, 20, 111]. These advantages have been recognized in machine learning as well, with grid-based representations increasingly integrated into transformers [112] and actor-critic agents [113] to improve task performance, efficiency, and interpretability.

Future work should examine how grid-based recurrent architectures solve the ForageWorld task—probing what spatial computations emerge and how they differ from standard RNNs. More broadly, our behavioral-neural analysis framework could be applied to other agent architectures and tasks, or used to generate hypotheses about navigation and memory in biological systems.

# G  Craftax licensing permissions

