# OpenReview forum: "Deep RL Needs Deep Behavior Analysis: Exploring Implicit Planning by Model-Free Agents in Open-Ended Environments"
_NeurIPS.cc/2025/Conference — NeurIPS 2025 poster_

### Official Review · Reviewer_se1j · 2025-06-21

**Clarity:** 2
**Significance:** 3
**Originality:** 3
**Rating:** 4
**Confidence:** 4

**Summary:**

In this paper the authors propose 1) a methodology for Reinforcement Learning agent analysis that goes beyond the inspection of reward curves and uses techniques inspired from neuroscience and  ethology to lend insight to agent performance and capabilities and 2) then aim to show that these analyses tools can be applied to a simple RNN based RL agent demonstrating that such an agent possesses some abilities, such as planning, that go beyond what was previously assumed in model free agents.

The authors make the following contributions:

 * Introduce ForageWorld, an environment that simulates ecological systems  for the study of planning and memory agents
 * Demonstrate how to use tools from Neuroscience and ethology (animal behaviour) for analysis and interpretability of RL agents.
 * Show that RNN models, trained in a model free fashion, can learn to plan and use memory.
 * Release of agent and environment code as open source.

The paper focuses on a set of analysis methodologies that depend on observing jointly the agent's internal representations and behaviour: Goal Inference, Memory Span, Planning Horizon, Spatial Structure, Network Capacity, and Representations.  It is further claimed that RL agents often fail to generalise well in open-ended environments and that these methodologies will provide a clearer indication on which models possess the requisite skills for such environments.

To study the efficacy of these methods a recurrent network (RNN) based agent trained with PPO with an auxiliary loss on position prediction in the grid along with some methods for introducing sparsity in the network. The agent's hidden state can also be decoded to reveal allocentric positioning throughout acting.

It is reported that through the use of the analysis methodologies it is determined that the agent explores effectively using revisitation strategies, structures its objectives over the course of an episode utilising its memory, exhibits the emergence of abilities over training, uses its state to support memory and planning, and supports generalisation across environment treatments.

**Questions:**

Line 28: I assume ML refers to "Machine Learning".  Referencing the term in full may add some clarity.

Does ForageWorld support multi-agent?

In the paper there is a lot of emphasis on agents trained via Deep RL methods but is this a necessary condition for the methods proposed?  Couldn't this be applied to agents optimised in different ways?  (e.g. behavioural cloning)

In the abstract, discussion, and possibly a few other times you indicate that there is a view that effective planning and memory use may require world models.  Is there some evidence that this view  is generally held? I was under the impression that model free agents can be fully capable of subsuming such capabilities.  Generative agent based models have certainly seemed to do so without any specific world modelling.

Why did you choose an RNN based agent?  Why not try feed forward as a baseline? What about a transformer based model?  Pre-trained models?

In the second paragraph of section 4.2 you refer to two figures (2 and 10), is this intentional?

**Ethical Concerns:**

["NO or VERY MINOR ethics concerns only"]

**Final Justification:**

This work presents a novel environment and a methodology for carrying out analysis on RNN based agents that lend concrete insights to biological intelligence.  Overall some of the main contributions include:

 * The ForageWorld environment which yields a naturalistic environment that is amenable to carrying out behavioural and neural analysis of agents.
   * Analysis is done on structured exploration, adoption of multi-objective policies, and the emergence of behavioural competencies of RNN based agents in ForageWorld.
 * Model free deep RL agents can successfully learn complex behaviour in naturalistic environments without explicit world models.
 * Argues that and provides evidence that this methodology can be used for neuro-science style analysis.

**Limitations:**

yes

**Paper Formatting Concerns:**

no.

**Quality:**

3

**Strengths And Weaknesses:**

**Strengths**

The work is well motivated toward agent analysis providing a comprehensive breakdown of behavioural competencies (Table 2).  Understanding agent capabilities is critical as agents deploy into more sophisticated and real world settings.

Draws inspiration from biology and neuroscience providing links into biological cognitive behavioural patterns and making use of existing and battle tested methodologies from those domains for understanding how neural activity is jointly represented with behaviour.

This approach highlights the importance of the interpretability of agents using methods that can gain insight into RL agents (or agents full stop). It is critical to understand agent capabilities especially as large models are increasingly deployed in the world. This paper presents a potential good step in the direction of contrasting and comparing such agents to understand which fundamental capabilities they have and will potentially provide more (or additional) insights than is done via agent evaluation benchmarks.

The authors demonstrate how a simple agent develops emergent planning and navigation strategies that are well adapted to the environment and generalise across episodes as the agent learns and retains information from its input for later use in an open-ended environment.

**Weaknesses**

I first want to clarify the main purpose of the paper.  It seems to me that the main takeaway is meant to be a novel suite of analysis methods inspired by behavioral and neural inspection methods from neuroscience etc.   There also seems to be a lot of emphasis on the capacity of the model free RNN based agent trained via PPO, is the primary purpose of this choice so as to showcase the efficacy of the analysis methods or is there some claim being made about this  specific agent architecture?  Further, there's some discussion about the sufficiency of neural analysis methods as applied to deep neural networks (as opposed to biological brains), it's unclear if this claim is addressed directly though the results in the paper, although the validity of the approach I think stands independently of this.  Overall, I think adding a bit more clarity about where its main contributions lie will be critical.

I found the overall presentation to be somewhat disorganised throughout the paper. Figures are referenced out of order and many referenced figures are in the appendix. Also, the conclusions drawn in the results section could be better backed up with empirical evidence from the analyses methods used and more clearly linked to their biological counterparts as claimed.  For instance, when discussing revisitation strategies done by the agent and how it develops its objectives throughout an episode it seems clear that the agent has learned these capabilities successfully however, it isn't completely clear how the methods were applied to draw these conclusions. Would it be possible to reference the targets in table 2 and how they were applied as part of an overall methodology?  Further to this point, I believe that I'd have a hard time reproducing your results from what's been put down in the paper. It may be that the appendix adds some clarification here but this type of stuff should be accessible in the main body of the work.  If the methods are the real takeaway of this work then I believe that their use and novel character should be made clearer.

The ForageWorld environment appears to be a relatively simple toy environment.  Do these results generalise well to other environments?  Are there limits to these analysis methods that depend on this? It would be nice to see whether the thesis about the interpretability of planning and memory capabilities holds up under more complex dynamics.  A few more details about this environment in the main paper and what motivated its choice would be very helpful.  If possible, considering an additional, more natural environment could lend further weight to this work.

It would be helpful to see a bit more justification around the model choice along with some additional detail around the architecture, training, RL algorithm and optimisation settings.  Further, some contrast to models that vary in some of these dimensions would be useful as a point of comparison especially from the perspective of the agent abilities that are discoverable through your analysis methods. Finally, like the environment, the choice of agent is somewhat limited compared to what's commonly used vis-a-vis large pre-trained models that can power generative agent based models. Considering such models would help ground the relevance of this work as well as provide a rich set of agent capabilities on which to perform this analysis.

---

> ### Author Rebuttal · Authors · 2025-07-31
>
> Thank you for your thoughtful comments! We will respond to specific concerns below:
>
> ### 1 (main purpose of the paper)
>
> We will clarify the language of how we describe our contributions in the manuscript, thank you for pointing out that this could be unclear. Our four main contributions are as you have listed in the summary. The core goal of this work is to demonstrate how studying standard deep RL agents in complex naturalistic environments using analysis methods from neuroscience can reveal complex behaviors and neural patterns that are informative for computational neuroscience and ML research.
>
> Further, we hope to encourage both computer scientists and computational neuroscientists to dig deeper into the joint behavior and neural data of agents in open-ended environments, as they provide controlled frameworks to test analysis methods intended for more complex real world biological data where controllability and statistical rigor may be lacking. Existing benchmarks do not easily provide the behavior, neural data, and naturalistic structure to be suitable for this, and thus we have provided both a new task and an analysis framework for it. Navigational circuits in large environments are especially poorly understood in both computer and biological sciences, as most well-studied tasks are small and/or fully observable arenas where complex planning with memory is not required. Thus, for novelty and impact we have focused on navigation in our analysis here, but many other aspects of an agent could be studied using similar techniques as well.
>
> We will add further references to the intro and expand discussion to contextualize this in the paper, such as:
>
> Abney, Drew H., et al. "Advancing a temporal science of behavior." Trends in Cognitive Sciences (2025).
>
> ### 2 (paper organization)
>
> Given the space requirements, we tried to have a coherent and complete narrative in the main text while referencing extensive methodological details and less critical experiments in the appendix.
>
> To address your specific points, we will fix the figure reference order and revise to better connect our findings in the results section with the analysis methods described in detail in the appendix that were used to obtain them. Thank you for helping us to improve the clarity of the paper!
>
> For reproducibility, all analysis and task code are available via an anonymized github link present in the main text of the paper, along with basic documentation on how to reproduce our experiments. We are committed to maintaining the project github publicly post-review. If there are specific requests for improvement we would be happy to respond to them.
>
> ### 3 (task complexity)
>
> We designed ForageWorld to find a balance between task complexity, transparency to analysis, and biological plausibility. It is certainly the case that more complex deep RL tasks exist, but many such tasks (for example, Minecraft) are much more difficult to study and in particular more difficult to instrument for behavior analysis. In addition, the increased computational cost of experiments in such environments is prohibitive to some statistical analysis, and many computational neuroscientists in particular do not have access to compute resources sufficient to run experiments in such complex environments.
>
> Relative to common neuroscience tasks used to study foraging, e.g. n-armed bandit tasks which lack spatial or temporal components, ForageWorld pushes the complexity boundary on the neuro-side of neuroAI while still remaining tractable for neuroscience tools and neuroscientists to study. Very complex tasks would be difficult to analyze with the current toolboxes used in neuroscience, which were designed for tasks of low-to-intermediate complexity. Excluding such analyses and many potential users by working with a much more complex environment would decrease the impact of our work.
>
> Further, ForageWorld as it exists is not trivial to learn- our agents require billions of timesteps of experience to train to competency on the task, and Craftax, the benchmark which we build upon to develop ForageWorld, has not been solved by any agent or architecture to date.
>
> ### 4 (model choice and hyperparameters)
>
> Certainly, we are happy to provide clarity on our model design choices. We will revise the manuscript to make explicit why we use an RNN model as well. We use RNN models because such models can be studied readily with neuroscience analysis techniques and have lower complexity than transformer models while still remaining general-purpose learning agents with hundreds of thousands of free parameters, allowing us to strike a balance in terms of compute requirements, amenability to analysis, and model capability. Our baseline PPO-RNN model is largely identical to the best-performing model presented by Craftax, other than the addition of the path integration output head and changes to model hyperparameters. On the original Craftax task, this PPO-RNN model is only moderately worse than the best reported model on the Craftax benchmark, a much larger transformer-based model which would not be compatible with many neuroscience analysis techniques [*].
>
> [*] Dedieu, Antoine, et al. "Improving transformer world models for data-efficient rl." arXiv:2502.01591 (2025).
>
> While an LLM-based model might be able to learn faster in ForageWorld by using prior knowledge, our goal is not to maximize performance on this task, but to understand what an agent trained de-novo on the task learns, and to prove out analysis techniques for obtaining such understanding. Many of the neuroscience analysis techniques we use would not apply to such a model, and we would not be able to measure learned features such as forward/backward position encoding in the RNN (figure 5). In addition, as ForageWorld is not a language-based task translating between its symbolic observation space  and an LLM would not be straightforward, and would introduce questions of prompting that have no connection to neuroscience or neural analysis but which would greatly impact performance. Further, using a pre-trained model such as an LLM makes connecting interesting behaviors to environment/reward function features (such as the emergence of planning in agent’s activations as a result of our open-ended reward function) impossible, as it is unclear what behavior is induced by the environment and what is derived from prior knowledge in the pre-trained model. Lastly, densely recording and analysing network activations across tens of thousands of timesteps per episode and tens or hundreds of episodes per training run (as we have done here) for an LLM would be intractable in terms of computational and storage requirements for the majority of academic labs.
>
> Training and environment details can be found in Appendix A, and model and environment hyperparameters are listed and selected values motivated in Table 3 in the appendix.
>
> ### 5 (multi-agent support)
>
> ForageWorld currently does not support multi-agent training, but we have ongoing work seeking to develop a multi-agent successor. Craftax also is not multi-agent.
>
> ### 6 (focus on deep RL training)
>
> None of our analysis is specific to deep RL training in particular, and we hope that future researchers interested in our work try many other types of training algorithms. We use deep RL here because we are interested in what emergent capabilities and features an agent will learn from training on a relatively simple reward function (maintain high resource levels and avoid death), and because such RL algorithms combined with memory networks (like RNNs) are of particular interest to neuroscientists as the best known computational approximation of animal learning. However, given a pre-existing behavioral cloning dataset (which, as ForageWorld is a novel task, did not exist prior to developing our deep RL agent) it is entirely possible to apply similar analysis to an agent trained via BC, or to any other agent with an RNN-type model. We will note these potential extensions in our discussion of future work, thank you for the suggestion!
>
> ### 7 (necessity of world models for planning)
>
> This view (that world models are necessary for explicit planning) is often stated in neuroscience and adjacent computational biology fields, although as written in the Bush et al (2025) paper [*] suggested by reviewer KWpH this topic is understudied considerably even in ML/AI (see the KpWH response about that contemporaneous work, which had a simpler task, shorter time horizons, full observability, and a reward structure that directly incentivizes planning). We will clarify this in the paper. On the neuro side, it was only recently appreciated that insects are individually quite intelligent, rather than just being swarm-governed or simple sensory-feedback automatons (see “Lars Chittka. The mind of a bee. Princeton University Press, 2022.” which we cited). The sort of planning a small insect brain can do is currently unknown, but their brains are only 1-2 orders of magnitude larger than our small DRL agent models, and the commonly-held historical view is that a brain of such size does not have the capacity to learn explicit planning without a world model (the existence of which has not been discovered by experimental neuroscience to date).
>
> [*] Bush et al. Interpreting Emergent Planning in Model-Free Reinforcement Learning. ICLR 2025.
>
> ### 8 (Feed-forward comparison)
>
> We did compare our RNN model to a feedforward agent, trained identically via PPO but with the RNN removed and replaced by an MLP of roughly similar parameter count. This ablation could not learn the task, as shown in Figure 2. This demonstrates that memory is required to learn our task to a non-trivial level of competency.
>
> ### 9 (Referring to two figures in section 4.2)
>
> Yes this is intentional. Figure 2 refers to the performance of feedforward (memoryless) agents quantitatively, while Figure 10 shows an example of the suboptimal pathing of such an agent.

---

> ### Comment · Reviewer_se1j · 2025-08-06
> **Rebuttal Response**
>
> Thanks for these clarifications, I've gone through your rebuttal and the other author's critiques and I'm happy to raise my score primarily given the following points from your rebuttal:
>
>  * The ForageWorld environment contains sufficient complexity while remaining accessible to neuroscience analysis.
>    * In particular, revisiting some of the analysis in the paper on structured exploration, adoption of multi-objective policies, and the emergence of behavioural competencies.
>  * Clarification of the claim involving World Modelling and the assumptions made in the neuroscience literature.
>  * Your commitment to address issues around paper clarity.
>
> These points address many of the issues that I had with the quality, novelty, and overall clarity of the work, thanks for clearing this up.  I still believe that it could be helpful to expand a bit on on your empirical findings regarding the contingency of world models in deep RL agents in naturalistic environments. You cover this a bit in the discussion section however it may be helpful to provide some insight as to: 1. whether there are conditions when a world model would be necessary for forage world type tasks, 2. the trade-offs to having a world model for the agent vs. the model-free approach taken, and 3. in what ways might a model-based agent reduce the clarity of the neural analysis.

---

> > ### Author Response · Authors · 2025-08-07
> >
> > Thank you for your thorough consideration, and for raising the score!
> >
> > Regarding your further questions:
> >
> > **1.** It is difficult to say when a world model is definitely needed or not needed (theoretically both will converge given infinite time), but based on prior work [1] environment features such as partial observability, the need for long horizon planning, goal emergence and selection, and multistep chains of actions, should benefit from world modeling, whether explicit or implicit. As we have some element of all these features in ForageWorld, we hope that it will be a useful benchmark for comparative analysis to help answer this question in future work.
> >
> > **2.** In general, the trend in [world] model-based versus model-free RL is that model-based methods are more data efficient to train but usually struggle to match the final performance of model-free algorithms, typically due to worse performance in regions of the state-action space where the model is less accurate. Whether this is a fundamental tradeoff or simply an artifact of existing algorithm families having technical limitations is unclear in the literature to our knowledge. As a result, model-based RL is typically more popular in robotics and related applications where data is scarce and expensive to collect and this data efficiency advantage matters more than peak performance, but this benefit is less critical for ForageWorld where we are not constrained by data availability.
> >
> > **3.** For our purposes the main challenge for studying world models is that interpretable world models (given present analysis methods) are often relatively low capability, while more capable world models are harder to analyze and understand (see [2] for additional discussion on this). Given that we observe evidence for learned world modeling in our model-free agent on a relatively complex task, this is strong motivation for future work to explore how the planning we observe differs from planning using an explicit world model, which might provide insights for how to improve world model based planning or model-based algorithms in turn.
> >
> > We will include consideration of these questions in the discussion section of the paper as well.
> >
> > Thank you again for your interest and engagement with our work- we agree there are a lot of interesting questions that can be asked to build off of our findings here, and we hope that we and others will be able to pursue many of them. If you have additional questions or thoughts we would be happy to answer during the remaining discussion period.
> >
> > [1] Dedieu, Antoine, et al. "Improving transformer world models for data-efficient RL." arXiv preprint arXiv:2502.01591 (2025).
> >
> > [2] Bush et al. "Interpreting Emergent Planning in Model-Free Reinforcement Learning." ICLR 2025.

---

### Official Review · Reviewer_KWpH · 2025-06-28

**Clarity:** 3
**Significance:** 3
**Originality:** 3
**Rating:** 5
**Confidence:** 3

**Summary:**

This paper introduces ForageWorld, a novel open-ended foraging environment inspired by real-world animal behaviors (e.g. depleting resources, predators, partial observability). The authors trained RNN-based, model-free DRL agents (PPO + GRU) to survive in this environment, and after which applied a behavior-first analysis toolkit adapted from neuroscience, cognitive science and ethology and find that trained model-free DRL agents exhibit structured foraging strategies with distinct phases and displays planning-like behavior without explicit planning modules or world models. The authors argue that by conducting joint behavioral and neural analysis for DRL agents, similar to those for animals, yields deeper mechanistic insight into how complex decision-making tasks are solved. Overall, this paper bridges ML and neuroscience by improving interpretability and evaluation of deep RL agents in complex environments with neuroscience insights.

**Questions:**

1. As mentioned in Weaknesses 1, a recent work [1] causally shows mechanistic evidence that model-free RL agents can exhibit planning-like behaviors. Could you clarify and summarize how this paper advances beyond [1]?
2. In Sec. 6.1 of [1], the authors inject learned “plan-concept” vectors to the agent’s activations and observe the causal effect of steering the agent from a short to a long route. Do you think there could be an analogous intervention, such as adding a “visit-patch-X” or “evade-predator” vector, that could demonstrate causal control over your agent’s planning-like behavior?
3. Can you provide several examples of candidate neuroscience tools, aside from those used in this paper’s experiments, that could be applied to evaluate DRL agents in ForageWorld?

**Ethical Concerns:**

["NO or VERY MINOR ethics concerns only"]

**Final Justification:**

While I disagree with the authors' description of Bush et al. as "contemporaneous work", after reading the rebuttal, I believe that this work is sufficiently different from Bush et al. and its results can be viewed as a further and nontrivial generalization of the finding of Bush et al. In addition, I appreciate the paper’s connection to neuroscience analysis tools, which is an aspect that distinguishes it from Bush et al. and bridges DRL and neuroAI. Therefore, I am leaning more towards accept than borderline accept and increase the score to 5.

**Limitations:**

yes

**Paper Formatting Concerns:**

No paper formatting concerns

**Quality:**

3

**Strengths And Weaknesses:**

**Strengths**
1. This paper adapted a behavior-first toolkit from neuroscience, cognitive science, and ethology (GLMs, phase segmentation, and linear state decoding) on the trained RNN-based DRL agents. Their analyses expose clear strategy phases (exploration → targeted patch revisitation) and show that allocentric position is linearly decodable up to roughly 50–100 steps into both past and future, demonstrating that planning-like computation can arise from purely model-free RNN agents alone.
2. Results of this paper draw a connection between DRL agent and animal behavior: For example, in Sec. 4.1, it states that agent's behavior phase shifts are analogous to animal foraging behavior phase shifts, and that their early trajectories resemble the “azimuthally rotating, outward spiraling loops” seen in ants and bees. In Fig. 3, it shows patch-choice GLMs rely on multiple variables such as proximity, predator history and uncertainty, which are commonly used to model rodent foraging decisions. By mapping agent strategies onto these established animal behavioral patterns, the work creates a concrete connection between DRL agent evaluation and behavior neuroscience, which may inspire future neuroAI work and interpretable RL evaluation.
3. This paper presents ForageWorld, a novel open-ended naturalistic foraging environment built based on Craftax by adding patchy renewable resources, dynamic predators and a survival-oriented reward function. This simulated environment provides a valuable tool for running complex, naturalistic behavioral experiments under realistic ecological constraints and cognitive demands. It can also serve as a platform for testing brain analysis techniques as the current DRL agents are likely simpler than biological agents.

**Weaknesses**
1. The discussion on existing model-free RNN-based DRL agent work is limited. As the authors claimed, one of the key contributions is showing “model-free RNN-based DRL agents can exhibit structured, planning-like behavior purely through emergent dynamics without requiring explicit memory modules or world models”. Although the authors briefly mentioned several earlier related literatures in Line 269-270, it did not discuss [1], a more recent paper that already claims to provide causal evidence of emergent planning in model-free RL agents. A detailed discussion will help clarify this paper’s novel contribution on this topic.

2. Current neural analysis is limited as it focuses on linear decoders of the GRU hidden state. One possible next step could be incorporating methods such as dynamical-systems analysis or representational-similarity analysis to uncover low-dimensional structure in the hidden activity. Those methods, which are widely used in neuroscience, could offer deeper insights into the trained GRU network.

[1] Bush et al. Interpreting Emergent Planning in Model-Free Reinforcement Learning. ICLR 2025.

---

> ### Author Rebuttal · Authors · 2025-07-31
>
> Thank you for your helpful comments! To respond to specific questions:
>
> ### 1 (comparison to Bush et al, and causal interventions)
>
> Thank you for bringing this work to our attention- we missed this recent paper during preparation, but it is definitely an important point of comparison, and we will discuss it specifically in the manuscript. It is also an interesting paper and was a thought-provoking read, the pointer is much appreciated.
>
> However, we must point out the conference rules on contemporaneous work:
>
> *“For the purpose of the reviewing process, papers that appeared online after March 1st, 2025 will generally be considered "contemporaneous" in the sense that the submission will not be rejected on the basis of the comparison to contemporaneous work. Authors are still expected to cite and discuss contemporaneous work and perform empirical comparisons to the degree feasible. Any paper that influenced the submission is considered prior work and must be cited and discussed as such. Submissions that are very similar to contemporaneous work will undergo additional scrutiny to prevent cases of plagiarism and missing credit to prior work.”*
>
> **Bush et al, with an Arxiv date of April 2nd, is considered contemporaneous work and thus a decision regarding our paper should not be based on comparison to this very recent paper.**
>
> That said, while Bush et al also show evidence of learned planning without a world model, ForageWorld differs from Sokoban in multiple important ways that make the planning evidence we see significant in excess of this prior work.
>
> **First**, ForageWorld is partially observable and not deterministic- our agent can only see a small portion of the arena (9x7 grid within a 96x96 arena, versus the 8x8 grids of Sokoban), and predator and food spawns are random but geographically biased, meaning the agent must plan under uncertainty and unknown information, which add significant barriers to the emergence of planning within the RNN model.
>
> **Second**, ForageWorld requires planning on a much longer timescale than Sokoban, with episodes lasting thousands of timesteps rather than dozens, meaning agents must plan and re-plan continuously within an episode rather than plan once and then execute.
>
> **Third**, our reward function is open-ended (keep food/drink/health/energy levels high), and does not directly reward planning over short time horizons, whereas Sokoban provides short-term feedback if planning is not successful via failed levels. When trying to survive for longer periods of thousands of timesteps our agent seems to plan out its exploration and revisitation of food patches due to the dynamics of patch depletion, and potentially maps out potential routes to evade predators. These behaviors are emergent, and are only indirectly rewarded across long time horizons of hundreds or thousands of timesteps.
>
> **Lastly**, our agent model and learning algorithm are very standard- we use a GRU with input/output fully-connected layers and PPO to train, as they are very commonly used in other research. This means that our finding of planning behavior is likely to generalize easily to other tasks/domains, and specialized models or algorithms are not required for planning to emerge. For example, the ability to perform multiple timesteps of computation per environment timestep is not essential for planning to emerge, per our results, which generalizes the findings of Bush et al.
>
> In terms of doing a similar causal intervention as Bush et al, it’s conceptually possible, but our environment is larger and more complex than the small fully observable 8x8 grid world used in [1]. Their intervention worked because they were able to handcraft a simple environment to have two possible solutions, and could trigger the agent to pick one over the other. Our open-ended survival task does not have such a way to binarize two “solutions” to a given arena configuration, as there are many patches, paths, and movement strategies an agent could take. Scaling their approach to an environment of our complexity would be very interesting, but is not trivial. Methods to intervene in agents doing long-term planning in complex environments in order to modify or correct their plans would make an interesting topic for future research.
>
> [1] Bush et al. Interpreting Emergent Planning in Model-Free Reinforcement Learning. ICLR 2025.
>
>
> ### 2 (additional analysis techniques)
>
> We agree that there are a wider range of neuroscience tools that could be applied to ForageWorld in follow-up work. While there are many possible options, to expand the immediate toolkit in the current work, we have added analysis and code (to our anonymized github linked in the paper) for using generalized linear models (GLMs) to predict single GRU neural spiking activity from behavioral variables for each GRU neuron. This addition allows us to better understand the functional circuits that exist within our trained networks. While we are not allowed to provide experimental plots here due to the recent conference guideline emails, this analysis reveals single neuron-level behavioral variable encoding that complements the current decoder results.
>
> The new neural-focused GLM experiments that we performed add evidence for an accumulation threshold circuit that pressures the agent to return to the origin the farther away it travels radially from the central start position. Specifically, we found the average response differential of position-selective neurons (i.e. the magnitude of the GLM regressor for a given position bin) increases with radial distance from the origin as the agent explores. This circuit could underlie the frequent origin revisitation we see our agents perform, with some sort of attractor driving the agent back to the origin in a ramping way with distance. Our GLM code is flexible, and can support single neural analyses using behavior time series that future researchers may come up with on our task or related ones, provided they use similar recurrent architectures and behavior logging. We have uploaded the code for this analysis to the anonymous github we linked in the manuscript and will add this analysis to the paper appendix.
>
> Representational similarity analysis would be difficult because it is designed for a lower number of simpler input states and usually time averages neural activity (the neural GLM predicts the full activity time series in contrast). Our agent has 3 different time-varying physiological levels and highly varying pixel input each time step, thus the conditions one contrasts with RSA and how time series are segmented would have to be carefully considered. We will add a note about this to the paper.

---

> ### Comment · Reviewer_KWpH · 2025-08-04
>
> Thank you for your thoughtful response!
>
> 1. I would like to point out that the Bush et al. is first published on 23 Jan 2025 on OpenReview as ICLR 2025 accepted paper. Therefore, unfortunately it does not fall within the definition of contemporaneous work. However, after considering the authors' rebuttal, I do agree with the authors that there is a considerable amount of difference between this paper and Bush et al. While this paper is not the first to demonstrate that RNN-based model-free DRL agents can exhibit planning-like behavior purely through emergent dynamics, it does consider this problem in a more difficult and general setting, thereby generalizing such finding of planning-like behavior.
>
> 2. The connection to neuroscience & ethology is an aspect that distinguishes this paper from Bush et al. and bridges DRL with neuroAI. The new neural-focused GLM experiment the authors introduced in the rebuttal further strengthens such connection.
>
> Based on the above points, I am leaning more towards accept than borderline accept and will increase the score to 5.

---

> > ### Author Response · Authors · 2025-08-04
> >
> > Thank you for thoughtfully considering our rebuttal and raising your score!
> >
> > If you have any other questions or suggestions during the discussion period we would be happy to receive and respond to them as well.

---

### Official Review · Reviewer_msZZ · 2025-07-01

**Clarity:** 3
**Significance:** 2
**Originality:** 2
**Rating:** 4
**Confidence:** 2

**Summary:**

The authors of the paper "Deep RL Needs Deep Behavior Analysis: Exploring Implicit Planning by Model-Free Agents in Open-Ended Environments" investigates how model-free reinforcement learning agents internally develop planning-like behaviors. To this end, the authors introduce an environment ForageWorld, which serves as an open-ended task inspired by animal foraging. Based on this the authors investigate properties of the learned GRU policy.

**Questions:**

* What environment features contribute to the emerging behaviors? I believe this study would benefit from ablations on the environment itself.
* Can the same findings also be obtained for pre-trained LLMs, since those are generally found to plan several tokens ahead too?

**Ethical Concerns:**

["NO or VERY MINOR ethics concerns only"]

**Final Justification:**

I believe the work is interesting and useful in that a highly optimized environment is presented that allows to easily study neural network agents on Neuroscience-relevant tasks.

**Limitations:**

* The study focuses on GRUs trained with PPO only. This is definitely a limitation as it does not support the broad applicability of the analysis pipeline.
* Unless otherwise convinced, I believe that environments such as Minecraft already capture many aspects of the proposed ForageWorld.
* It is unclear which environmental aspects cause specific behaviors in the agent

**Paper Formatting Concerns:**

There are no formatting concerns.

**Quality:**

3

**Strengths And Weaknesses:**

Strengths:
* ForageWorld exhibits rich dynamics with ecological motifs, which overall provides a nice testing ground.
* The efforts of this paper are well motivated and follow an interdisciplinary approach.
* It is notable to find that models such as GRUs that are trained model-free to exhibit planning-like behavior.

Weaknesses:

* The authors study only a single type of agent, which is a GRU trained with PPO, which does not speak for the broad applicability of the methods employed in this work. In particular, one would have hoped that the authors study also LLM-based agents (without or with fine-tuning) on such a task. This would not only strengthen the broader applicability point, but also serve to make the work more visible.
* There is little discussion or evidence surrounding this environment's advantage as compared to other open-ended environments such as Minecraft (which is mentioned). Could one not perform the same analysis on such environments?

---

> ### Author Rebuttal · Authors · 2025-07-31
>
> Thank you for your comments. To respond to specific concerns:
>
> ### 1 (why not an LLM agent)
>
> While an LLM-based model might be able to learn faster in ForageWorld by using prior knowledge, our goal is not to maximize performance on this task, but to understand what an agent trained de-novo on the task learns, and to prove out analysis techniques for obtaining such understanding. Many of the neuroscience analysis techniques we use would not apply to such a model, and we would not be able to measure learned features such as forward/backward position encoding in the RNN (figure 5). In addition, as ForageWorld is not a language-based task translating between its symbolic observation space and categorical action space and an LLM or VLM would not be straightforward, and would introduce questions of prompting that have no connection to neuroscience or neural analysis but which would greatly impact performance. Further, using a pre-trained model such as an LLM makes connecting interesting behaviors to environment/reward function features (such as the emergence of planning in agent’s activations as a result of our open-ended reward function) impossible, as it is unclear what behavior is induced by the environment and what is derived from prior knowledge in the pre-trained model. Lastly, densely recording and analysing network activations across tens of thousands of timesteps per episode and tens or hundreds of episodes per training run (as we have done here) for an LLM would be intractable in terms of computational and storage requirements for the majority of academic labs.
>
> Using LLM architectures for computations underlying foraging, navigation, search & rescue, etc. tasks is an interesting future direction, but considerable work needs to be done on developing analysis techniques suitable for those architectures first. We will cite preliminary work in this direction for the future work appendix section, such as:
>
> Qiao, Yanyuan, et al. "Open-nav: Exploring zero-shot vision-and-language navigation in continuous environment with open-source llms." arXiv preprint arXiv:2409.18794 (2024).
>
>
> ### 2 (why not use Minecraft)
>
> While Minecraft represents a more challenging procedurally generated survival environment, and studying the behavior and neural activity of RL agents in it would be worthwhile research, there are a number of factors that would make the contributions of our work difficult or impossible to obtain using Minecraft.
>
> **First**, instrumenting Minecraft to log the range of environment variables that we record would be impossible without assistance from Microsoft to modify the source code/API, which would make it impossible to perform many of our experiments (for example, relating patch revisitation rates to specific environmental factors, as shown in figure 3). This similarly limits the extendability of ForageWorld to modify factors such as arena generation or predator type/behavior.
>
> **Second**, Minecraft’s game logic runs on CPUs, whereas ForageWorld is implemented in JAX and runs completely on GPU, allowing for massive parallelization in rollouts within a single compute node and training for billions of environment timesteps, which was necessary to observe the de-novo emergence of planning, for example. Performing a similar scale of experiment in Minecraft would require vastly more computational resources and software infrastructure to do massively distributed rollouts across a large compute cluster, and would greatly limit the number and duration of experiments that could be run. We also note that such compute resources are out of reach for many research groups, especially computational neuroscientists, which would limit the impact of our environment and analysis tools in follow-on work. This computational accessibility coupled with high task complexity was a major feature of Craftax as a framework to build upon, as it allows future work to easily build upon our work (as we have done with regard to Craftax).
>
> **Third**, while doubtless many interesting phenomena could be observed in Minecraft, none of our main contributions would be significantly enhanced by using Minecraft- a more complex environment is not needed to observe foraging with patch revisitation in a long-time-horizon POMDP and the de-novo emergence of planning, nor for adapting neuroscience analysis tools to deep RL agents.
>
> **Lastly**, we are interested in analyzing agent behavior and neural dynamics when trained from scratch on an open-ended foraging environment, and the complexity of Minecraft environments would make training such an RL agent to do sophisticated temporally-extended foraging involving predator evasion and patch revisitation a formidable challenge technically and scientifically, as to our knowledge prior work demonstrating complex behavior in Minecraft has relied heavily on pre-training on human data (though if there is any recent work training complex tasks from scratch in Minecraft we would be curious to learn about it and study the methods used).
>
> ### 3 (environment features that influence behavior)
>
> This is a very interesting question, which we explore along several dimensions in the paper. We explore how factors such as arena size, agent field of view, and the presence/absence of the position prediction auxiliary loss term affect behavior (figure 2). We found that each of these factors produce changes in agent behavior and/or dynamics which are not obvious from comparison of reward curves. In future work there are of course more factors that could be varied to explore their influence (for example, a multi-agent version of the environment, including learned predators), and we hope to explore such extensions to the core foraging task in future work.
>
> ### 4 (additional model(s) for comparison)
>
> We focused on a single standard DRL architecture because our focus was on developing and demonstrating the environment and analysis techniques, and on demonstrating that planning can emerge without explicit world modelling, not to argue that such emergence is a common feature of many algorithms/models, as making such a claim would demand a much wider range of environments and models to compare. For further comparison, we implemented and tested an additional DRL-RNN model, PQN [*] with an LSTM (changing both RL algorithm and model architecture). We found PQN-RNN models can perform the task comparably well to our PPO models, with similar behavior and learning curves. However, interestingly, relative to PPO agents the PQN agents seem to have higher position prediction performance, prefer curvier paths on short time scales, and are worse at killing predators (prioritizing predator evasion rather than the killing predators with tools as PPO learns to prefer). We will summarize these updates and provide the code in the updated manuscript.
>
> [*] Gallici, Matteo, et al. "Simplifying deep temporal difference learning." arXiv preprint arXiv:2407.04811 (2024).

---

> > ### Comment · Area_Chair_Szeu · 2025-08-06
> >
> > Dear reviewer msZZ,
> >
> > Could you please respond to authors' rebuttal as soon as possible?
> >
> > Thank you!
> > AC

---

> > ### Comment · Reviewer_msZZ · 2025-08-06
> >
> > I would like to thank the authors for their very detailed response and clarifications. The authors convinced me about my mentioned weaknesses (to some degree), and I would like to encourage them to more prominently mention ForageWorld's GPU acceleration, which I missed.
> >
> > I do believe that it would be really interesting to investigate pre-trained LLMs (already with agentic capabilities) with the same methods presented here, which is almost certainly possible and would not require much resources. I do, however, acknowledge, that this is beyond the scope of this work.
> >
> > Hence, I am willing to raise my score correspondingly.

---

> > > ### Author Response · Authors · 2025-08-07
> > >
> > > Thank you for your thoughtful response and consideration, we greatly appreciate raising the score!
> > >
> > > We will be sure to highlight the GPU-accelerated nature of the benchmark going forward, this is good feedback. We agree that investigating LLM behavior on ForageWorld would be interesting and informative as a comparison, though there are a number of technical questions to solve in doing so, and it's something we are hoping to contrast with in future work using ForageWorld to compare behavior and dynamics across agent types/models.

---

### Official Review · Reviewer_bGpG · 2025-07-03

**Clarity:** 4
**Significance:** 3
**Originality:** 3
**Rating:** 4
**Confidence:** 4

**Summary:**

This paper introduces “ForageWorld”, a procedurally generated open-ended reinforcement learning environment with ecological structure  (built on an existing framework called Craftax). The environment replicates ecological pressures through partial observability, diminishing food sources, shifting predator threats, and biological drives like hunger and fatigue. They conduct joint behavioral and neural analysis using tools adapted from neuroscience, cognitive science and ethology to study reinforcement learning agents trained in ForageWorld environment (using path analysis, generalized linear models, and recurrent state decoding). The presented work demonstrates that analyzing models this way reveals subtle learning strategies and internal representations that standard performance measures often overlook, and shows connections between artificial and biological systems.

The authors train PPO agents across five configurations (predator blind agent - agent that cannot cause damage to the predators, normal field of view - aerial view of shape 9x11 grid around the agent with the agent in the center, front-facing field of view - an aerial view of the grid rows in-front of the agent instead of 9x11 grid with agent the center as input, auxiliary path integration objective - auxiliary loss function for linear decoding of the position in reference to the start position as the origin from the recurrent state) -

1. Baseline RNN agent (non predator-blind agent with normal field of view and auxiliary path integration objective)
2. Predator-blind RNN agent
3. RNN agent without auxiliary path integration objective
4. RNN agent with front facing field of view
5. Memoryless agent (feed-forward neural network instead of RNNs)

- The authors highlight the emergence of structured phase like foraging behavior analogous to those observed in animals. Similar to rodents, after initial exploration of the environment, the models show goal-directed revisitation of previously discovered states. Additionally, the authors suggest that the initial exploration qualitatively resembles insect search behavior and show that agents integrate information in a multi-objective manner while making revisitation decisions.
- In recurrent state decoding analysis the authors find that the past and future allocentric positions (position with respect to the origin/start-state) can be decoded from the recurrent state of the agent better than the egocentric position (position with respect to the agent’s current position), and that the path integration objective is necessary for decodable spatial representations. In addition, decoding accuracy improved over training and removing path integration loss reduced performance on large arenas due to agent’s inability to predict its own position.
- Pruning for sparse activations improved allocentric decoding in RNN agents.
- Memoryless agents show reduced survival and poor coverage of the arena in comparison to the RNN agents. They do not exhibit emergent phase like foraging behavior and get trapped in local loops.
- Both predator-blind agents and non predator-blind agents achieve similar survival performance and converge to similar movement strategies favoring predator avoidance instead of combat.
- Agents with front-facing field of view performed more dense local exploration in early exploration period, and transitioned faster to longer range exploration compared to normal field of view.

**Questions:**

- It is unclear how path 1, 2, 3 and 4 were classified in Figure 1, 6, 10 and 14. Do they belong to different episodes or the same episode? If they belong to the same episode, what was the criteria of differentiation?
- How does survival time evolve across episodes during the training process?
- What dependencies or correlations exist between survival time and other experimental metrics throughout training?
- In Figure 4 and similar figures, the 'distance traveled' subplot contains 'Early in Episode' and 'Late in Episode' categories, but it is unclear how these categories were defined. Additionally, the predator related metrics could be better explained. The subplot uses the title “pred. kill” whereas the text mentions “predator defeat”. Is it possible to provide more comprehensive definitions for the metrics used in this figure?
- The caption for Figure 11 states that "The absence of structured spatial uncertainty plots (top left) reflects degraded internal spatial representation," but this is unclear. How does the absence of the plot indicate degraded spatial representation?

**Ethical Concerns:**

["NO or VERY MINOR ethics concerns only"]

**Final Justification:**

Issues resolved - methodology clarity, literature coverage, and figure explanations. (I have updated my rating for clarity)

Remaining concern - Reward function sensitivity analysis not addressed - limits understanding of behavior robustness vs. design artifacts. The authors show the emergence of animal behavior, but do not explain in detail how/why these behaviors emerge in relation to the structure of the environment (or) reward.

Overall, it is interesting work with compelling emergent behaviors and rigorous analysis. The unresolved reward sensitivity issue is outweighed by the methodology and clear research value, which makes me maintain my overall rating.

**Limitations:**

yes

**Paper Formatting Concerns:**

I found no significant problems or concerns.

**Quality:**

4

**Strengths And Weaknesses:**

### Strengths -

I would like to thank the authors for submitting their interesting work.

- Their work introduces a new benchmark with ecological constraints useful for studying emergent phenomena in deep RL agents which could be of interest across multiple disciplines.
- The emergent agent behavior observed in this work and its similarities to animal behavior is very interesting.
- The experimental analysis clearly isolates and demonstrates the emergent phenomena discussed by the authors.
- The comprehensive evaluation across multiple agent configurations strengthens the validity of the conclusions.



### Weaknesses -

- While agent behavior in reinforcement learning is significantly influenced by the reward function, the paper does not investigate how variations in reward design affect emergent behaviors.
- The experiment for agents with front-facing field of view seems incomplete (Figure 15) as it contains only early-to-mid training data.
- A more comprehensive review of related work on emergent cognitive behaviors and interpretability in reinforcement learning would strengthen the paper's positioning.
- A more detailed explanation of the metrics in Figure 4 would enhance the paper's clarity.

---

> ### Author Rebuttal · Authors · 2025-07-31
>
> Thank you for your insightful comments! To address the specific questions:
>
> ### 1 (variations in reward design)
>
> This is an interesting question, thank you for asking it. We did experiment with several different reward function variants during development of ForageWorld (for example, instantaneous reward for eating when hungry), but did not explore them in detail because the resulting behavior was degenerate/not foraging-like in various ways (the instantaneous reward agent for example would try to be on the edge of hunger as much as possible to maximize reward from eating). As our goal was to develop and study agents performing simulated foraging behavior, we did not discuss these variations in the paper since they do not lead to naturalistic behavior and there are many subtle environment design details that can break naturalistic foraging (for example, the spawn rate of cows relative to the food value per cow must be set carefully to prevent either unavoidable starvation or the agent managing to survive indefinitely in one patch). We will add a brief description of some of these variants to the appendix.
>
>
> ### 2 (incomplete figure 15 learning history plot)
>
> Thank you for catching that- we had actually standardized on the shorter run durations for most experiments (as the major behavior and dynamics features seem to be established within that duration), but in hindsight that makes the directional vision experiment look out of place in that figure compared to the longer runs. We performed longer runs to show the full time series for that experiment and have amended our manuscript (the trend and conclusions are consistent with the original results).
>
>
> ### 3 (related work on emergent cognitive behaviors and interpretability)
>
> This is a good suggestion- we will enhance our literature review to include work on that topic, including:
>
> Abney, Drew H., et al. "Advancing a temporal science of behavior." Trends in Cognitive Sciences (2025).
>
> Verma, Abhinav, et al. "Programmatically interpretable reinforcement learning." International conference on machine learning. PMLR, 2018.
>
> Cheng, Zelei, Jiahao Yu, and Xinyu Xing. "A survey on explainable deep reinforcement learning." arXiv preprint arXiv:2502.06869 (2025).
>
> ### 4 (expanded definitions for metrics used in figure 4)
>
> We apologize if we were unclear about the calculation of metrics that were plotted across the agent’s training period. We will provide more detailed definitions of each metric in the appendix and refer to it. The numbers given are the average per episode value of the metric at a given training time point. For example, predator kills is the mean number of predators killed in one episode (i.e. in one arena configuration). Food score is the mean value of how satiated the agent is in a given episode, with a score of 9 as zero hunger and 0 as max hunger. Predator “kill” and “defeat” are indeed the same phenomenon (reduced the predator HP to zero with repeated attack actions), we will relabel these captions to be consistent. Thank you for your help in clarifying the manuscript!
>
> ### 5 (paths used in analysis)
>
> We thank the reviewer for pointing out this was unclear. Indeed, the paths here belong to the same episode, and are directly consecutive from 1 through 4 in the behavioral time series logs for the episode. Path 1 starts at t=0 of the episode. This is also true for the related appendix figures which show additional representative examples. The path differentiation is done manually for this figure with the purpose of illustrating how a representative agent frequently opens new episodes by doing azimuthally rotating loops as an exploration strategy. We will add this explanation to enhance clarity.
>
> ### 6 (survival time across episodes)
>
> Survival time is closely correlated to total return, and thus we chose to only list returns. This follows from the reward function in section 3.1, which means an agent that survives for longer will very likely be spending more time above the 50% mark on its resource levels and thus receiving positive reward.
>
> ### 7 (relationship between survival time and other metrics)
>
> As can be seen by comparing the Fig. 2 performance plot and the Fig. 4 learning history plot, most behavioral metrics change with improvements in survival time at first, but plateau 30-50% of the way through training. Increase in exploration area correlates with early survival time improvements, but late training improvements seem to be driven by more efficient predator defense plus possibly additional factors that we could not identify, thus leaving an open question for future work. We will expand the current discussion to address this point.
>
> ### 8 (unclear Fig. 11 caption)
>
> We apologize that the caption was not written clearly. The top left plot is missing because there is no position predicting auxiliary objective in this experiment, thus no error in position prediction to report. The quoted sentence about the “absence of structured spatial uncertainty” was supposed to refer to Figures 2 and 17 which show further information about this experiment, where the worse decoder performance and worse survival performance in the absence of the position prediction suggests that internal spatial representations may be worse in the agents which do not predict their own position. We will improve the caption and reference those figures.
>
> ### 9 (clarifying the definition of late/early in figure 4 legends)
>
> Thank you for the comment and we apologize for the confusion. “Early in Episode” corresponds to the first 1500 time steps of a new episode and “Late in Episode” is after the first 1500 time steps of a new episode. We will clarify this in the paper.

---

> > ### Comment · Reviewer_bGpG · 2025-08-06
> >
> > Thank you for your thorough rebuttal addressing all my questions and concerns. I appreciate the detailed responses and the commitment to improving the manuscript. The planned improvements should address most of my concerns and enhance the paper's clarity.
> >
> > Your response (1) highlights a delicate balance required in designing ecologically realistic environments. While I understand that degenerate behaviors motivated focusing on the current reward structure, I would suggest conducting a sensitivity analysis to examine how emergent behaviors respond to perturbations in the reward function parameters as the reward structure seems crucial for foraging behavior and would inform future design choices.
> >
> > The work's contribution to understanding emergent behaviors in ecologically-structured RL environments through behavioral and neural analysis is valuable, and I maintain my positive assessment of the research.

---

### Decision · Program_Chairs · 2025-09-17

**Decision:**

Accept (poster)

**Comment:**

This paper presents a valuable study of emergent planning-like behavior in standard model-free deep RL agents trained in a novel open-ended environment, ForageWorld. The authors show that GRU-based agents trained with PPO can exhibit long-horizon behavior resembling planning, despite the absence of explicit memory or world models. They further support their claims using neuroscience-inspired tools to analyze internal representations and behavioral strategies.

The paper’s key strengths include the well-motivated task design, comprehensive behavioral analysis, and the use of interpretable neural probing techniques. The authors provide detailed implementation and analysis resources to strengthen reproducibility.

Reviewers initially raised concerns regarding novelty in analysis methodology and the lack of comparison to Bush et al. (2025). The authors highlighted important distinctions in task complexity and experimental design from Bush et al. They also provided code for using generalized linear models (GLMs) to predict single GRU neural spiking activity from behavioral variables for each GRU neuron. Furthermore, the authors provided detailed discussion on the limitations of applying alternative methods like RSA in their setting.

Overall, the authors have convincingly addressed all reviewer concerns during the rebuttal. The paper makes a solid and timely contribution to understanding the emergence of planning-like behavior in RL agents.